# A Climate Data Record of Sea Ice Age Using Lagrangian Advection of a Triangular Mesh

Anton Korosov[1], Léo Edel[1], Heather Regan[1], Thomas Lavergne[2], Signe Aaboe[3], and Emily Jane Down[3]

[1]Nansen Environmental and Remote Sensing Centre, Bergen, Norway
[2]Norwegian Meteorological Institute, Oslo, Norway
[3]Norwegian Meteorological Institute, Tromso, Norway

**Correspondence:** Anton Korosov (anton.korosov@nersc.no)

**Abstract.** We present LM-SIAge, a new Climate Data Record (CDR) of Arctic sea ice age spanning the period from 1991 through 2024. The dataset is based on a novel Lagrangian advection scheme applied to a triangular mesh, which conserves sea ice age fractions and reduces numerical diffusion compared to the previous Eulerian approach. LM-SIAge is derived from satellite observations of sea ice concentration and drift, and represents fractional age classes per grid cell. The record captures the spatial and temporal evolution of first- to sixth-year ice, including uncertainty estimates that account for both sea ice concentration and drift uncertainties.

We compare LM-SIAge with existing products from NSIDC and C3S, finding consistent large-scale trends—such as the decline of older ice—but also identifying systematic differences. Trend analysis confirms a significant reduction in sea ice age and a general increase in the area of first-year ice. Validation with ice drifting buoys indicates good consistency (LM-SIAge does not underestimate max age of the buoys in 98.3% of cases), with most discrepancies occurring near the ice edge. The NSIDC product does not underestimate the age in 96.4% of cases.

The LM-SIAge dataset improves the observational basis for Arctic monitoring and contributes to the Global Climate Observing System (GCOS) Essential Climate Variables. It is publicly available and suitable for climate studies, model evaluation, and data assimilation.

## 1 Introduction

Sea ice age is one of the key indicators of the state and evolution of the Arctic sea ice cover. Older ice tends to be thicker, more resilient to melting, and a better indicator of long-term climatic changes. Sea ice age is increasingly being used in reanalysis systems, for example, to correct biases in sea ice thickness reconstructions (Edel et al., 2025). It is also valuable for validating coupled model simulations of sea ice evolution (Regan et al., 2023). As sea ice roughness varies with ice age (Johnson et al., 2022), and since altimeter-derived thickness retrievals are sensitive to surface topography (i.e. roughness, Landy et al., 2020), incorporating sea ice age can substantially improve thickness estimates from altimeter data by constraining this uncertainty source. Sea ice age has become a standard attribute in modern ice charts, with older (multiyear) ice categories incorporated into Electronic Navigation Charts (ENCs) and Electronic Chart Display and Information Systems (ECDIS) to inform Arctic route planning and enhance navigational safety (Falkingham, 2025).

Recognising its significance, the Global Climate Observing System (GCOS) recently included sea ice age in the pool of essential climate variables (ECV) (Lavergne et al., 2022). However, long-term, high-resolution records of sea ice age remain limited, and the existing datasets (e.g., Fowler et al., 2004; Korosov et al., 2018; Tschudi et al., 2020) have limitations in terms of representation of age distributions or temporal coverage.

In this study, we present a new Climate Data Record (CDR) of Arctic sea ice age for the period 1991–2024. The dataset
is derived using a novel Lagrangian advection algorithm applied to a triangular mesh. Sea ice age fields are initialised each autumn, when the ice extent reaches its annual minimum and all remaining ice is assumed to be multi-year. The subsequent evolution of the ice age is computed by advecting this initial field using daily satellite-derived sea ice drift vectors, while accounting for deformation and melt. This approach enables us to reconstruct the continuous age distribution of sea ice in both space and time.

The dataset builds on our earlier work (Korosov et al., 2018), which used a similar conceptual approach but was limited to the years 2000–2012 and relied on an Eulerian advection scheme. That scheme introduced artificial diffusion, leading to overly smoothed ice age fields. In contrast, the new method employs a Lagrangian approach, where nodes of a triangular mesh are advected without numerical diffusion, preserving the sharp gradients and internal structure of the ice pack. This also enables tracking of multiple age fractions within each grid cell, in contrast to the widely used U.S. National Snow and Ice Data Center
(NSIDC) sea ice age product (Tschudi et al., 2020), which retains only the oldest age class per cell but spans 1981–2023.

The resulting dataset provides a consistent and physically based estimate of sea ice age distributions over more than three decades, offering new opportunities for analysing long-term changes in Arctic sea ice structure, dynamics, and resilience.

## 2   Data

Similar to the previous version of our algorithm, we use satellite-derived sea ice drift (SID) and sea ice concentration (SIC) as
the input data.

### 2.1   Sea Ice Drift data from EUMETSAT OSI SAF

For the period from 1991 through 2020 we use OSI-455, the EUMETSAT Ocean and Sea Ice Satellite Application Facilities (OSI SAF) global low-resolution sea ice drift climate data record (CDR) (OSI SAF, 2022c; Lavergne and Down, 2023). For 2021–2024, we utilise the Global Low Resolution Sea Ice Drift product, OSI-405-c, which is a near-real time (NRT)
product (OSI SAF, 2007; Lavergne et al., 2010). Both products utilise the same satellite sensors, SSM/I (>= F10) and SS-MIS (CMSAF FCDR), AMSR-E (NSIDC) and AMSR2 (JAXA). The NRT product also utilises C-band scatterometer data from the ASCAT missions. Both products employ the same sea ice motion-tracking methodology, continuous maximum cross-correlation (CMCC, Lavergne et al. (2010)), which involves a fractional-pixel pattern matching of the brightness temperatures. These brightness temperatures are pre-processed with Laplacian filters, and the vectors are then post-processed with correction
schemes.

The product configurations differ between the CDR and NRT products. The CDR utilises an EASE-2 75 km grid, whereas the NRT product is based on a Polar Stereographic grid of 62.5 km. There is also a mismatch between the periods over which the motion vectors are retrieved, with the CDR using a period of 24 hours and the NRT product using a period of 48 hours. The final difference is that, due to the difficulty of retrieving vectors directly from the satellite data in the summers during the pre-AMSR-E period, a summer gap-filling method is used for the CDR. The summer motion vectors of the CDR are retrieved from an implementation of the free-drift model (Thorndike and Colony, 1982; Thomas, 1999; Brunette et al., 2022), using the ERA5 wind reanalysis and trained on the motion vectors from running CMCC on AMSR-E (2002-2011) and AMSR2 (2012-) brightness temperatures.

The CDR and NRT products were intercompared by running the sea ice age algorithm for an overlapping period of three years, as shown in Section 4.4, and were found to be consistent enough for generating a continuous sea ice age CDR.

## 2.2 Sea Ice Concentration from EUMETSAT OSI SAF

The global sea ice concentration climate data record version 3 from SMMR/SSMI/SSMIS data (Lavergne et al., 2019; OSISAF, 2022b) was used for dates up to and including 2020, and the accompanying interim climate data record (ICDR) version 3 (OSI SAF, 2022a), based on SSMIS data, for dates 2021 – 2024. These products are both on a 25 km EASE2 grid (Brodzik and Knowles, 2011). These products are retrieved using the SICCI3LF algorithm based on the 19 GHz and 37 GHz imagery. Information about the land/water mask was obtained from the status flag of the OSI SAF SIC product.

All OSI SAF products can be accessed from the web portal at https://osi-saf.eumetsat.int/.

## 2.3 Pathfinder sea ice age from US NSIDC

The dataset produced in this study is compared to the existing ice age dataset from NSIDC and the ice type dataset from the EU Copernicus Climate Change Service (C3S).

The EASE-Grid Sea Ice Age, Version 4, was downloaded from the NSIDC portal (Tschudi et al., 2019) for the period 1991 - 2023. The product is generated at NSIDC by Lagrangian tracking of particles seeded in multi-year sea ice and advected using the Polar Pathfinder Daily 25 km EASE-Grid Sea Ice Motion Vectors, Version 4 (Tschudi et al., 2020). The advected particles are binned on daily intervals, and the oldest particle defines the age of sea ice in the grid cell of the product.

## 2.4 Sea Ice Type Climate Data Record from C3S

The sea-ice type CDR, version 4 (Aaboe et al., 2023a), downloaded from the C3S Climate Data Store (Aaboe et al., 2023b), is a daily classification product that maps the dominant ice types, first-year ice, multiyear ice, and an ambiguous ice class, across the Arctic at 25 km resolution. Here, multiyear ice is defined as all ice that has survived at least one summer melt, corresponding to a second-year ice or older. It is derived using a Bayesian classification algorithm applied to passive microwave brightness temperatures from the SMMR, SSM/I, and SSMIS (CM SAF FCDR), combined with atmospheric reanalysis data (ERA5) and auxiliary sea-ice information. The product employs a temperature-based correction scheme to minimise the misclassification

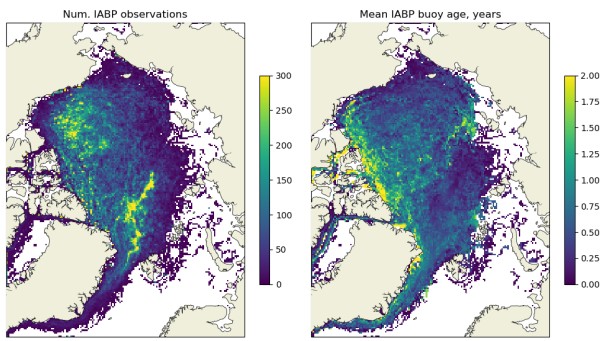

**Figure 1.** The map on the left shows the number of buoy positions collected between 1991 and 2024, and the map on the right shows the average age of the buoys.

of ice types in situations where warm air infiltrates sea-ice-covered regions. It also incorporates sea-ice drift information, both to correct misclassifications through a backtracking scheme and to refine the daily tuning of the algorithm. The ice-type classification is available only during the winter months (October to April) and covers the period from 1978 to the present. The product is provided on a 25 km EASE2 grid.

## 2.5 Ice drifting buoy trajectories

For validation, we used the same reference data as in (Lavergne and Down, 2023) acquired between 1991 and 2024. Figure 1 shows maps of the number of ice drifting buoy locations (left) and average age of a buoy (right) computed as discussed in Section 3.9.

## 2.6 Preprocessing of sea ice drift

The sea ice drift fields from the OSI SAF NRT product were reprojected and rotated to the CDR grid as described by Lavergne (2023) and converted from 2-day drift to 1-day drift fields by dividing the speed magnitude by 2. Daily fields were stacked into a 3D data cube and processed with a 3D Gaussian filter, with a kernel size of $0.5 \times 1 \times 1$ in the time-, x-, and y-directions, truncated to 2 standard deviations. Before smoothing, the gaps in ice drift fields (on land or on no-data areas) were filled using the nearest neighbour method. After smoothing, the extent of the ice drift fields was adjusted to match the extent of the SIC product (i.e., pixels with concentration above 15 %). Finally, missing drift files were created using linear interpolation from neighbouring files.

## 3 Methods

### 3.1 Overview of the sea ice age algorithm

Like our previous algorithm (Korosov et al., 2018), we compute sea ice age from ice concentration and drift in the following way. A pan-Arctic field of concentration is taken from satellite observations at the end of the melt season, when the ice extent is minimal, and all ice is assumed to be multi-year ice (MYI, $C_{MY}$) (Fig. 2, Step 1). The $C_{MY}$ field is repetitively advected (morphed) using daily satellite-observed sea ice drift fields (Fig. 2, Step 2; see video "advect_myi.gif" from the video supplement (Korosov, 2025a)). Changes of concentration due to ice deformation or melting are accounted for during the

advection process (see Section 3.3 below). Some time after the initialization of the MYI field advection (e.g. on 1 Jan 1992, as shown in Figure 2), the advected field represents concentration of MYI which is lower than the total observed concentration and is denoted as $C_{A0}$ (i.e., advected for less than one year). The difference between the total and advected fields yields the concentration of the first-year ice (Fig. 2, Step 3):

$$C_{1Y} = C_{TOT} - C_{A0} \tag{1}$$

Hereafter, we define the first-year ice ($C_{1Y}$) as ice formed during the ongoing freezing season and that has not yet experienced melting.

One year after the initialisation, the advected field (Fig. 2, Step 4) represents the concentration of sea ice which is at least two years old and is denoted $C_{A1}$ (advected for one year). At that time, the total observed concentration again reaches a minimum, representing the concentration of multi-year ice. $C_{MY}$ has higher values than the advected $C_{A1}$ as it also contains a fraction of

120 the second-year ice (Fig. 2, Step 5):

$$C_{2Y} = C_{MY} - C_{A1}. \tag{2}$$

Hereafter, we define the second-year ice ($C_{2Y}$) as ice formed during the previous freezing season and that has survived one melting season. It should be noted that according to the nomenclature of the World Meteorological Organisation (Sea Ice Nomenclature, WMO-259), the first-year ice (FYI) that survives the summer minimum is called "residual ice", and it turns into

125 second-year ice only on 1 January of the coming winter. Nevertheless, in this work we define that all survived FYI turns into the second year after 15 September.

As shown on Figure 3, both $C_{A1}$ and the current $C_{MY}$ are advected further using the sea ice drift and after one more year they become $C_{A2}$ and $C_{A1}$, i.e. ice fractions advected for two years and for one year. Since $C_{A1}$ contains $C_{A2}$ and, similar to the previous year, the new $C_{MY}$ contains $C_{A1}$ we can compute fractions of the second- and third-year ice:

$$C_{2Y} = C_{MY} - C_{A1}$$

$$C_{3Y} = C_{A1} - C_{A2} \tag{3}$$

This workflow is repeated, and the ice age fraction can be computed using a generic formula:

$$C_{NY} = C_{A(N-2)} - C_{A(N-1)}, \tag{4}$$

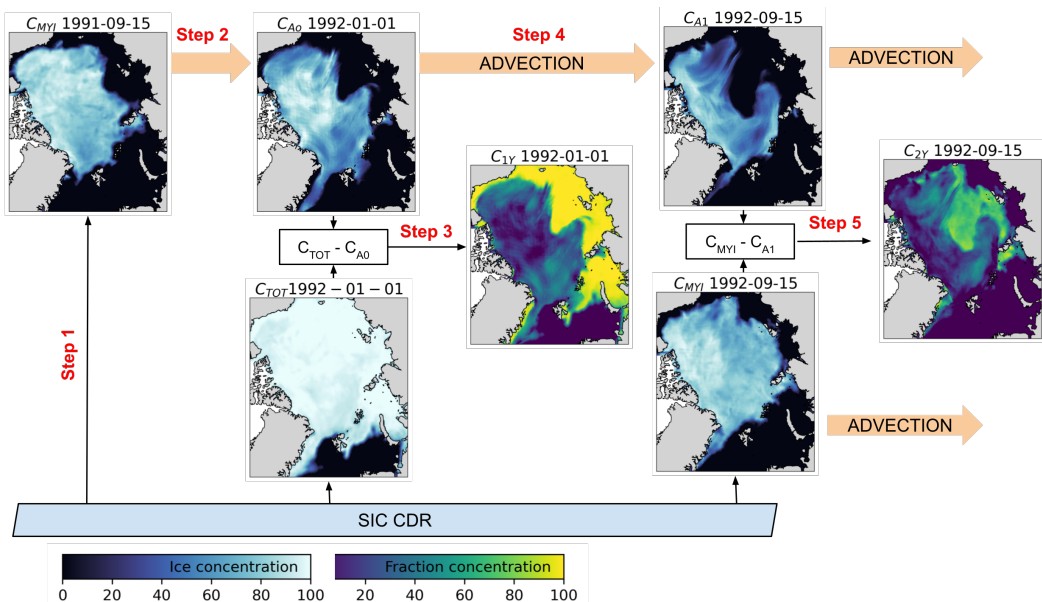

**Figure 2.** A scheme of sea ice age fraction computation within one year. The fields of MYI and TOT concentration originate from SIC CDR (Ice concentration colour map). These fields are advected using sea ice drift from the SID CDR (advection is shown by thick orange arrows). The advected fields of MYI are labelled as "$C_{Ai}$ YYYY-MM-DD", where $i$ denotes the number of years of advection and YYYY-MM-DD denotes the date of the advected field. MYI and advected fields are used to compute sea ice age fractions (Fraction concentration colourmap), labeled as "$C_{jY}$ YYYY-MM-DD", where $j$ denotes the age of the ice fraction, i.e. $C_{1Y}$ stand for the first-year ice and $C_{2Y}$ stands for the second-year ice.

where $N$ is an integer number indicating the age of sea ice fraction.

The primary difference in the new algorithm lies in the advection scheme. Previously, we used an Eulerian scheme, which
resulted in the diffusion of the ice fraction fields on each step and overly smoothed results. In the new algorithm, we use a Lagrangian scheme, where nodes of a triangular mesh are advected using the ice drift vectors. Triangular elements in this mesh are advected without diffusion unless a remeshing occurs (see videos "advect_disk.gif" and "advect_myi_zoom.gif" from the video supplement (Korosov, 2025a)). The sections below provide a detailed description of the advection scheme.

### 3.2 Generation of initial mesh

The initial triangular mesh is created for the region of interest, including the Arctic Ocean above 60°N and excluding the Baffin Bay and Canadian Archipelago, by triangulating points on a regular grid in EASE2 projection (Brodzik and Knowles, 2011) with ≈ 25 km spacing (see Figure 4). Points located on land more than 150 km from the coastline are excluded from triangulation. The mesh is optimised with the Laplace method using the GMSH library (Geuzaine and Remacle, 2009). The nodes of the mesh located on land are marked as fixed: they cannot be moved by the advection or remeshing procedures.

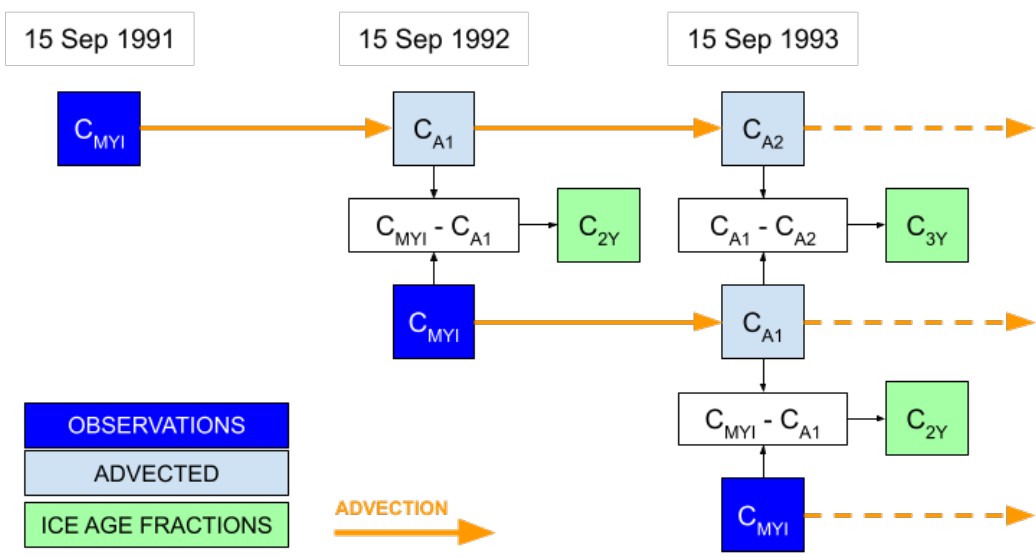

**Figure 3.** A scheme of sea ice age fraction computation for multiple years. The field of multi-year ice from observations is shown as dark blue blocks, advected fields are shown as light blue blocks, and the computed ice age fractions are shown as green blocks. Orange arrows show advection.

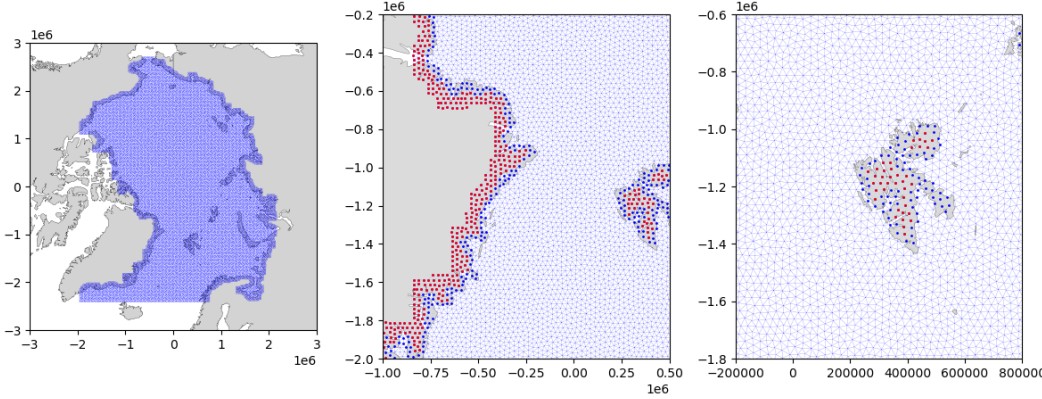

**Figure 4.** The initial triangular mesh is shown at three zoom levels. The red dots indicate the locations of nodes that cannot be moved during remeshing, and the blue dots show the locations of nodes where the interpolated ice drift was set to zero.

Nodes of the generated initial mesh are advected every day using daily sea ice drift satellite products, and the mesh is remeshed after each advection as described below.

### 3.3 Advection of mesh nodes and remeshing

The daily ice drift vectors from the OSI SAF product are linearly interpolated on the nodes of the triangular mesh and position of the nodes is updated: $X_{n+1} = X_n + U_n$, where $X_n$ is the initial position of nodes, $U_n$ is ice drift velocity in km d$^{-1}$ and $X_{n+1}$ is the new node position.

After the advection, some elements (triangles) of the mesh are critically distorted and require remeshing. The following criteria are set to detect distorted elements:

- Edge of the element is shorter than 13 or longer than 38 km;

- Element has an angle below $15°$;

- Element area is smaller than 20 km$^2$;

- Element is flipped.

The end results are not very sensitive to the mesh size. These parameter values are chosen to keep the area of the mesh elements below that of the destination grid elements with a spatial resolution of 25 km, while ensuring the elements are large enough for efficient advection and, especially, to avoid time-consuming remeshing.

The following operations are applied recursively to the mesh, changing one edge at a time (see Fig. 5 for illustration):

- If an edge is shorter than a threshold or an angle in the element is below a threshold, the shortest edge of the element is collapsed: two nodes that belong to the shortest edge are replaced with the one node between them, and the element that had the shortest node is removed (Fig. 5, A).

- If an edge is longer than a threshold, the edge is split in two: a node is added in the middle of the edge, and the initial large element is replaced with two smaller elements sharing one new edge (Fig. 5, B).

- If an element is flipped, the edge over which it is flipped is removed, and the created quadrangle is bisected by the edge connecting the two other nodes (Fig. 5, C).

After all defective edges of the mesh are updated, the elements that were changed and the other distorted elements (i.e., those with a small area or a small angle) are selected on the mesh, together with their neighbours and neighbours of neighbours. The selected elements are regularised with the Laplace method (Nealen et al., 2006) for generating a mesh with more uniform elements as illustrated in Figure 6.

### 3.4 Mapping between advected meshes

Most of the elements of the mesh at step $n+1$ are not distorted enough to require remeshing and have a corresponding element on the mesh at step $n$. For these elements, a conservative mapping is used – the entire content from the element on the previous step is transferred to the corresponding element on the next step:

$$A_{i,n+1} = A_{i,n}, \tag{5}$$

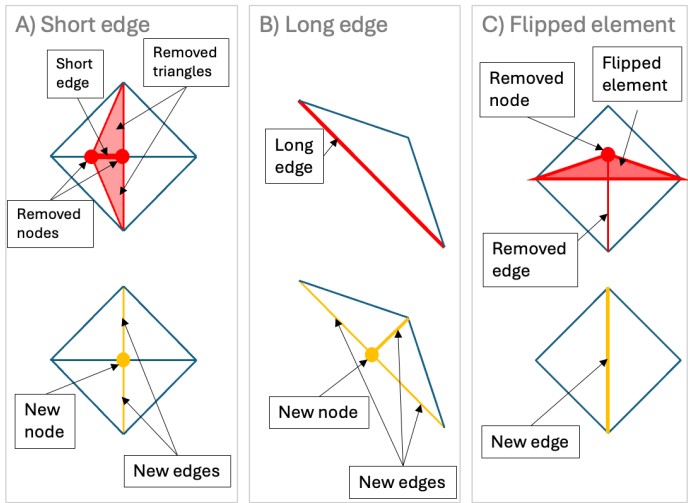

**Figure 5.** Scheme of three types of remeshing: collapsing of a short edge (A), splitting of a long edge (B), removing a flipped element (C). Edges and elements before remeshing are shown in red in the upper row, and the new mesh is shown in yellow in the lower row.

where $A$ is an areal (i.e., not voluminal) content, e.g., area of sea ice in an element.

Since the area of the non-remeshed element may change, the concentration ($C$) of an areal content (e.g., sea ice concentration) also changes:

$$C_{i,n+1} = k_i C_{i,n},\tag{6}$$

where $k_i$ is a factor equal to the ratio of the changed element area ($a_i$):

$$k_i = a_{i,n}/a_{i,n+1}\tag{7}$$

For the elements that were remeshed, we find all elements on the advected mesh that they intersect with (see Figure 7). The total areal content of the remeshed element is equal to the weighted average of the intersecting elements, where the weight is proportional to the area of intersection. The concentration of an areal content is, therefore, equal to the weighted average of concentrations adjusted by the factor accounting for change in the element area:

$$C_{i,n+1} = \frac{\sum_j (w_j k_{i,j} C_{i,j,n})}{\sum_j w_j},\tag{8}$$

where $w_j$ is the weight of the j-th intersecting element. Note that this equation is generic and can also be used for conservative mapping with $j \in [1]$ and $w_1 = 1$.

After each advection, the indices of the elements on the source mesh, along with their weights, are saved together with the new mesh to facilitate quick mapping between meshes. The advection and remeshing process described above is applied to the whole timeseries of sea ice drift data once to produce daily meshes and mappings between them. These meshes and mappings are then used to advect the sea ice concentration field.

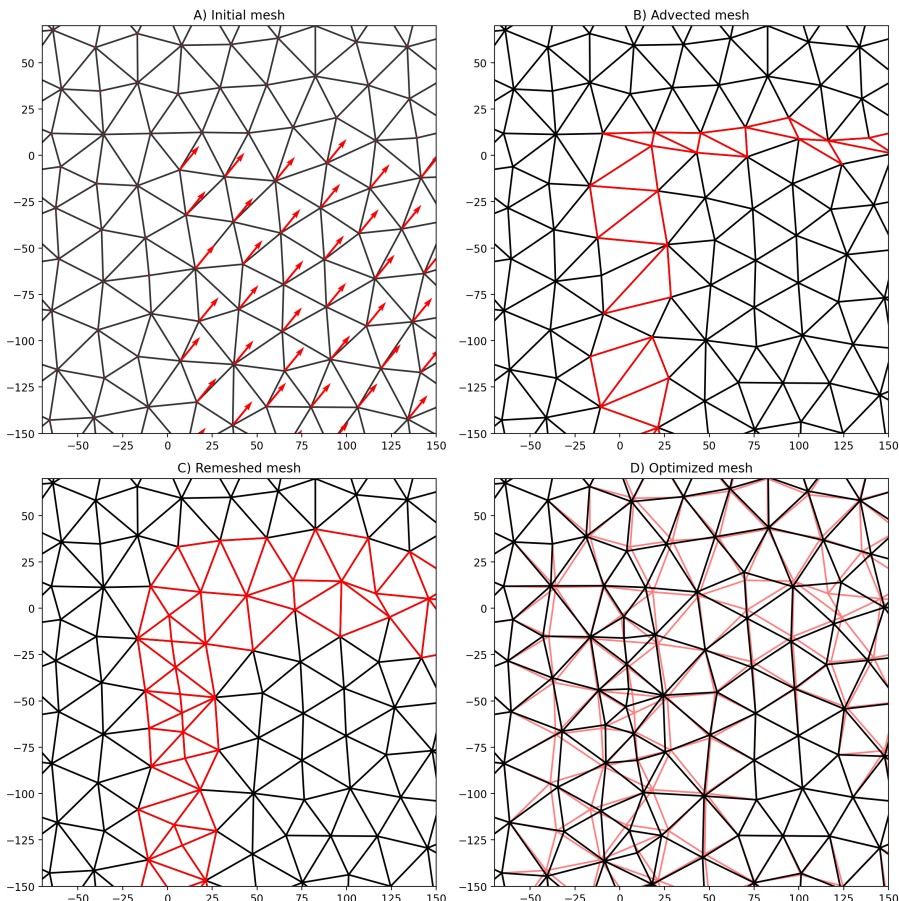

**Figure 6.** Illustration of the node advection and mesh optimisation process. A) Some nodes of the initial mesh (shown in black) are advected using ice drift vectors (shown in red). B) The advected mesh (shown in black) has some distorted elements that require remeshing (shown in red). C) Most of the elements on the remeshed mesh remain unchanged (shown in black). The new elements introduced by remeshing are shown in red. D) Position of the nodes in the remeshed mesh is updated, and a regularised mesh is created (shown in black). The previous mesh (remeshed, but not regularised, shown in red) differs from the optimised one only near the new elements, in the vicinity of the convergence/divergence zone. In contrast, in the homogeneous ice-drift area (lower right corner), the advected mesh is equal to the remeshed and optimised meshes.

## 3.5 Advection of sea ice concentration field

Prior to advection, the initial sea ice concentration field ($C_{OBS,n}$) for a selected date is linearly interpolated from the OSI SAF gridded product to the centres of elements of the corresponding mesh. For the next day, the corresponding mesh and the mapping are loaded, and the advected concentration ($C_{n+1}$) is computed using Eq. 8. The OSI SAF concentration for the next

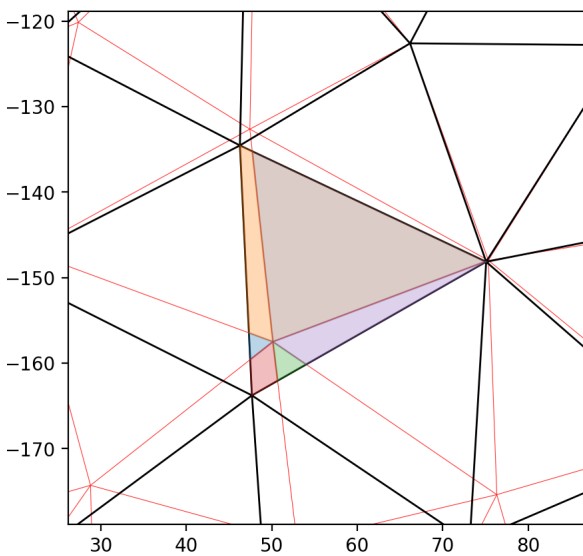

**Figure 7.** Illustration of computing weights for the intersecting mesh elements. The previous (advected) mesh is shown by red colour. The new (remeshed and optimised) mesh is shown in black. Intersections of the central element from the new mesh with elements from the previous mesh are shown by polygons of various colours. Their areas are used as weights in Eq. 8

day ($C_{OBS,n+1}$) is also interpolated on the same mesh. The observed concentration conditions the advected concentration:

$$\hat{C}_{n+1} = min(C_{n+1}, C_{OBS,n+1}). \tag{9}$$

Conditioning is necessary in cases where the element area decreases due to convergence, resulting in a concentration that exceeds the observed value. Conditioning can be interpreted as ice ridging – the excess of sea ice area in a compressed element increases the sea ice thickness, which we cannot account for, as we don't have accurate enough observations of sea ice thickness during the entire period of SIC and SID observations. The observed concentration can be lower than the advected one, also due to errors (e.g., an unaccounted atmospheric impact on brightness temperatures of passive microwave data). In such a case,
the advected concentrations are also conditioned, and the uncertainty of $C_{OBS}$ is used for computing the uncertainty of the advected field as described below.

### 3.6 Initialization of the minimum concentration field

In the previous version of the algorithm, the concentration of 15 September was assumed to be minimal and was taken as the concentration of multi-year ice. However, in the central part of the Arctic, freezing starts earlier, and by 15 September, the
first-year ice is already present in some regions.

To avoid including FYI in the MYI map, we modified the annual initialisation procedure as follows. We take the observed total concentration fields on all days between 5 September and 14 September and advect them until 15 September independently of each other without capping. Then, the minimum concentration from the advected fields is taken into the MYI concentration

field:

$$C_{MYI} = min(C_{A,i}), \tag{10}$$

where $C_{A,i}$ is the advected concentration starting from $i$-th date ($i$ ranging between 5 and 14 September).

### 3.7 Generation of the Lagrangian mesh sea ice age climate data record (LM-SIAge CDR)

We initialised the algorithm for mesh advection on 1 January 1991 and advected the mesh using SID CDR until 2020, and using the NRT SID product until 2024 as described in Sections 3.2 - 3.5. The MYI concentration field was computed for 15 September for years 1991 – 2024 as described in Section 3.6. Each MYI concentration field is advected for 2200 days (6 years). Thus for each day after 15 September 1995, we obtained 6 fields of $C_{A1} - C_{A5}$. For the spin-up period before 15 September 1995, we had fewer fields with advected MYI. Since the same meshes and mappings were used for advection of MYI concentration originating from different years, for each day, the $C_{A1} - C_{A5}$ fields are located on the same mesh, which facilitates computation of sea ice age fractions using Eq. 4. For dates between the minimum ice extent, the first-year ice concentration was also computed using Eq. 1.

The sea ice age is then computed using a weighted average as suggested in our previous algorithm:

$$A_a = \frac{\sum_i (A_i C_{iY})}{\sum_i C_{iY}}, \tag{11}$$

where $A_i \in [1, 2, 3, 4, 5, 6, 7]$ is an integer year, $C_{iY}$ is the fraction of ice of corresponding age ($C_{1Y}$ corresponds to FYI).

Values of the ice age fractions, as well as the average ice age, were linearly interpolated on a regular grid in the EASE2 projection with a resolution of 25 km, matching the grid of the OSI SAF SIC CDR, and exported to netCDF files.

### 3.8 Computation of uncertainties

The uncertainty of the sea ice age variable (Eq. 11) is computed by propagation of uncertainties in the input SID and SIC products, as shown on the flowchart in Figure 8. It starts from computing the uncertainty of the advected sea ice concentration field $C_{n+1}$ from Eq. 8:

$$\sigma_n^2 = k_n \sigma_{n-1}^2 \tag{12}$$

where $n$ denotes the current step, $n-1$ - the previous step, and $k_n$ denotes the factor for change of area (and, therefore, uncertainty) due to divergence/convergence (see Eq. 7).

Since the observed concentration conditions the concentration in the advected field (i.e., it is a minimum value of the advected and the observed concentration (see Eq. 9), the uncertainty of the conditioned field is the uncertainty of the minimal concentration:

$$\sigma_{MIN,n}^2 = \begin{cases} \sigma_{n-1}^2, \text{ if } C_{n-1} < C_{OBS,n} \\ \sigma_{OBS,n}^2 \end{cases} \tag{13}$$

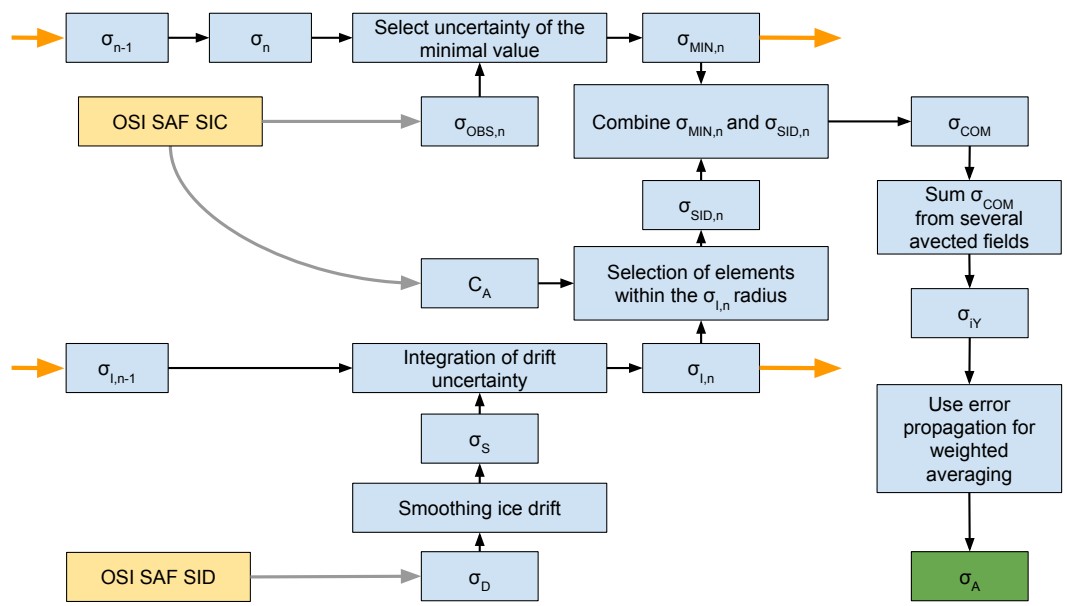

**Figure 8.** Flowchart of computing the uncertainty of sea ice age. The input data is shown in yellow, and the final result is displayed in green. The orange arrows indicate data flow using the advected mesh. See Eqs. 12 - 19 for the notation of individual uncertainty components.

This uncertainty of the advected and conditioned concentration ($\sigma_{MIN}$, Eq. 13) is then used to compute the combined uncertainty of the advected concentration field:

$$\sigma_{COM}^2 = \sigma_{MIN}^2 + \sigma_{SID}^2, \tag{14}$$

where $\sigma_{SID}$ is the uncertainty in concentration associated with uncertainty in ice drift (see details below in Eqs. 15 - 17).

Computation of $\sigma_{SID}$ starts from computing the uncertainty of the smoothed ice drift product ($\sigma_S$) using the provided uncertainties of the OSI SAF SID product ($\sigma_D$):

$$\sigma_S^2 = \frac{\sum_i^N \sigma_{D_i}^2}{N}, \tag{15}$$

where $\sigma_{D_i}$ is the uncertainty of gridded ice drift in a $i$-th neighbour and $N$ is the number of neighbours.

Next, the integrated uncertainty of the ice drift ($\sigma_I$) is iteratively accumulated with advection:

$$\sigma_{I,n}^2 = \left\langle \sigma_{I,n-1}^2 \right\rangle + \sigma_S^2 \tag{16}$$

After some steps of advection, the integrated uncertainty of sea ice drift defines a radius of a circle, where the advected mesh element could have drifted. This circle may include other advected elements with their respective concentrations. Therefore, the uncertainty of advected MYI concentration, associated with the uncertainty of ice drift $\sigma_{SID}$, is computed as a standard

deviation of concentrations in this circle:

$$\sigma_{SID}^2 = \left( \sum_j^M C_{A,j} - \langle C_A \rangle \right)^2 / M, \tag{17}$$

where $j$ is the index of the elements in the circle, $\langle C_A \rangle$ is the average concentration in the circle, and $M$ is the number of elements in the circle.

After the combined uncertainty of several undevcted MYI fields is computed (Eq. 14), we continue with the uncertainties of ice age fractions. Each ice age fraction is calculated as the difference between advected MYI fields (Eq. 4); therefore, their uncertainty is computed from the combined uncertainties of advected MYI fields:

$$\sigma_{NYI}^2 = \sigma_{COM,N-2}^2 + \sigma_{COM,N-1}^2, \tag{18}$$

where $N$ is an integer number indicating the age of sea ice fraction.

Finally, the uncertainty of the produced sea ice age variable ($\sigma_A$) is computed from the uncertainties of the ice age fractions. The error propagation formula in the case of weighted averaging reads as follows:

$$\left( \frac{\sigma_q}{q} \right)^2 = \left( \frac{\sigma_x}{x} \right)^2 + \left( \frac{\sigma_y}{y} \right)^2 \tag{19}$$

where

$$
\begin{aligned}
q &= x/y \\
x &= \sum_{i=0}^{N} A_i C_{iY} \\
y &= \sum_{i=0}^{N} C_{iY} \\
\sigma_x^2 &= \sum_{i=0}^{N} A_i * \sigma_{C_{iY}}^2 \\
\sigma_y^2 &= \sum_{i=0}^{N} \sigma_{C_{iY}}^2
\end{aligned}
\tag{20}
$$

where $C_i$ is concentration of $i$-th ice age fraction multiplied by its age ($A_i$), $i$ is the index of the sea ice age fraction: $i = 1$ for one-year-old ice (i.e., $C_{1Y}$), $i = 2$ for two-year-old ice, etc.

## 3.9 Validation

For validation of the LM-SIAge and NSIDC products, we used trajectories of sea ice drifting buoys. We compare the maximum ice age detected by the Lagrangian algorithms to the age of the buoys. These two quantities are not expected to match one-to-one because we do not know the age of the ice on which the buoy was deployed. Instead, the validation tests the hypothesis that the age of the buoy cannot exceed the maximum ice age detected by our LM-SIAge. The maximum ice age from the LM-SIAge is computed as the maximum age of the ice fraction above 15 %. The age of the ice-drifting buoy is calculated from the time of its deployment. The buoy trajectories are collocated with the OSI SAF SIC product to detect the melting and freezing of ice around the buoy. If a buoy is deployed in open water or ends up in open water (OSI SAF concentration drops below 15 %) during summer melt and then refreezes again into sea ice (the concentration grows above 15 %), the age counting is reset to 0.

 **4   Results and discussion**

## 4.1   LM-SIAge dataset description

Figures 9 and 10 show maps of sea ice age fractions and associated uncertainties for 31 December 1995 and 31 December 2024 from the LM-SIAge dataset. In both years, FYI concentration is low in the central part of the Arctic and the Canadian sector and is high in the eastern and peripheral seas. Fractions of older ice occupy the central part, with the oldest ones being closer to the Canadian archipelago. Unlike 2024, concentrations of fourth-, fifth- and sixth-year ice are still relatively high in 1995. SIC uncertainty is quite heterogeneous, with higher values in the middle of the ice pack and near the sea ice edge. It increases from $0-5$ % for FYI to $3-10$ % for the older ice age fractions. SID uncertainty is zero for FYI, as ice drift is not used for advection. For the older ice age fractions, SID uncertainty increases from 5–10 km to 300 km, primarily in the Beaufort Gyre, where the resilient time is extended, or in the Fram Strait, where velocities are high. Due to the higher ice drift speed in 2024, the SID uncertainty is significantly higher (100–600 km). SIC uncertainty associated with the SID uncertainty is zero for FYI. It increases from $0-5$ % to $20-25$ % for older ice fractions, mainly at the edges of advected MYI fields, where spatial variations in MYI concentration are high. The total uncertainty is dominated by the SIC uncertainty for FYI and second-year ice fractions ($\approx 5$ %) and grows up to 25 % for older fractions due to the impact of the SIC_SID uncertainty.

## 4.2   Seasonal and interannual variations of uncertainty

We analysed the variability of average uncertainty in the source data and in the derived dataset (see Fig. 11). The observed SIC uncertainty ($\sigma_{OBS}$, Fig. 11, A) shows strong seasonal variations, with a minimum ($\approx 2$% ) in winter and a maximum ($\approx 4.5$%) during the melt season. The observed SID uncertainty ($\sigma_S$, Fig. 11, B) also has a minimum ($\approx 3.5$ km d$^{-1}$) in winter, a plateau of constant values of 4 km d$^{-1}$ in summer, and two peaks with $\approx 7$ km d$^{-1}$ just before and after the summer period. The uncertainty of the advected MYI field ($\sigma_{MIN}$, Fig. 11, C) starts from $\approx 3$ % and gradually decreases over 6 years, with a slight increase during summer seasons. The integrated uncertainty of ice drift ($\sigma_I$, Fig. 11, D) starts from nearly zero and increases step-wise following the pre- and post-summer peaks of $\sigma_S$. The uncertainty of advected MYI concentration, associated with the uncertainty of ice drift ($\sigma_{SID}$, Fig. 11, E), also begins low and then rapidly increases during the first year, which is followed by a gradual increase during consecutive years with substantial seasonal variations. The total uncertainty of the advected MYI field ($\sigma_{COM}$, Fig. 11, E) is first dominated by the uncertainty in the observed SIC, but after the end of the melt season and the jump of $\sigma_S$, the contribution of $\sigma_{MIN}$ becomes much less pronounced.

## 4.3   Comparison of LM-SIAge, NSIDC and SIType dataset

Figure 12 shows the average and maximum sea ice age for 31 December for every 5 years computed from the LM-SIAge dataset and compared to the NSIDC sea ice age and the SIType CDR products. In the first year, LM-SIAge is still in the spin-up period, and the maximum age is underestimated. For other years, the maximum age from the LM-SIAge product shows good correspondence to the NSIDC dataset and a similar extent of the MYI as in the SIType CDR product. In both the LM-SIAge

and NSIDC products, a gradual decrease in ice age can be observed; however, the LM-SIAge product provides a more detailed view of the fate of individual ice age fractions. Video supplement (Korosov, 2025b) includes an animation of the average sea ice age for the entire dataset.

One minor difference between the LM-SIAge and NSIDC products, which is difficult to spot, is the presence of MYI near the coast in the Kara and Laptev Seas (also visible in the supplementary videos). This is not realistic and results from enhanced sea ice concentrations near the coast in the upstream SIC products due to the "land spillover" effect (Kern et al., 2022). These pixels are masked in the netCDF files.

Figure 13 shows sea ice area by age class derived from LM-SIAge (the first column for each year), from the NSIDC product (second column for each year, except 2024) and the SIType CDR. For LM-SIAge, the pixel area was multiplied by
the corresponding sea ice age fraction and then summed up. In the NSIDC and SIType CDR, individual pixels were summed up for each ice class. By this, we expect the NSIDC and SIType to show equal or higher areas by class than the LM-SIAge since they do not include fractional pixel area in their computation as LM-SIAge does. All products agree in showing a general decline of older ice, which is especially pronounced in 2007, when MYI extent was at a minimum and in 2012, when almost all ice older than 4 years disappeared. Despite using the same projection and mask for computing ice fraction areas from different
products, systematic biases appear to exist between the products. NSIDC seem to underestimate the total concentration before 2011 and then overestimate it compared to the LM-SIAge. Total ice area and MYI area are also overestimated by the SIType product compared to the LM-SIAge and NSIDC.

We estimated linear trends in the change of area of ice age fractions for the three products over the period from 1995 through 2024 as shown in Figure 14. The rate of FYI increase is relatively consistent for LM-SIAge, NSIDC, and SITYpe CDR (40,000
330  km$^2$ y$^{-1}$, 42,000 km$^2$ y$^{-1}$, 55,000 km$^2$ y$^{-1}$, correspondingly). In contrast, a decrease in MYI is consistent only for LM-SIAge and SIType CDR ($\approx$-60,000 km$^2$ y$^{-1}$) and is underestimated by NSIDC (-33,000 km$^2$ y$^{-1}$). Changes in the second-year ice are not significant, and older ice categories are losing area at a rate of approximately -10,000 km$^2$ y$^{-1}$, as observed in both the LM-SIAge and NSIDC products.

We also computed linear trends in the reduction of sea ice age for the same products as in Figure 12, i.e., the weighted
average age of LM-SIAge, the maximum age of LM-SIAge, and the NSIDC. The trends were estimated in every pixel using values from a $3 \times 3$ pixel sliding window (see Figure 15) for 31 December of years from 1995 (when the LM-SIAge product was not in spinup) through 2024. The average age product shows a weak decline of MYI age (-0.58 month y$^{-1}$). In contrast, the max-age products show a stronger "rejuvenation" of MYI: -1.43 and -1.92 month y$^{-1}$ for LM-SIAge and NSIDC, respectively. It's also interesting to note the slight spatial difference between LM-SIAge and NSIDC products, despite the similarities in age
distribution in earlier maps: the NSIDC main losses do not extend as far west and remain closer to the Canadian Archipelago.

Several factors lead to the discrepancies observed between the LM-SIAge, NSIDC and SIType products. Firstly, the LM-SIAge and NSIDC are derived from MYI advection, whereas SIType is a radiometric product. Next, LM-SIAge and NSIDC use quite different ice drift products, advection schemes, and representations of the ice age state. In addition, LM-SIAge provides the MYI concentration for each pixel. Summing the areal coverage of MYI, weighted by its concentration, gives the total MYI
area. In contrast, NSIDC and SIType products provide a categorical classification, assigning a fixed ice-age class to each pixel.

In these cases, the total MYI coverage is obtained by summing the areas of all pixels classified as MYI, which corresponds more closely to an MYI extent. As a result, the ice extent is generally larger than the ice area.

Finally, we don't account for the different convergence (and melting) rates of ice of various ages, as we cannot constrain these rates by observations. Assuming that older ice is thicker and can converge (melt) less than the thinner younger fractions, we may overestimate the loss of older ice in converging (melting) cells. Our previous experiments with a numerical model-based estimate of sea ice age (Regan et al., 2023) indicate that ridging younger ice first (but melting at the same rate) yields more realistic estimates of MYI extent. However, that may lead to an underestimation of MYI ridging in areas where MYI and FYI thicknesses are similar (e.g., the marginal ice zone) and in recent years, as MYI thins out faster than FYI (Kwok, 2018).

### 4.4 Comparison of CDR and NRT SID products

To evaluate whether we can use the NRT SID product after 2020 for computing the ice age CDR, we performed an experiment where we advected a field of MYI concentration from 2017 using the CDR and NRT products for three years, from 1 January 2018, to 31 December 2020. Figure 16 compares the maps and total area of the advected fields and shows only minor differences. The spatial distribution of the MYI advected using the NRT product appears slightly smoother, and its area is almost equal to that of MYI advected with the SID CDR. Only towards the end of the test period, during the third melt season, and only when the advected concentration field gets low, the area difference reaches 15 %. We can therefore conclude that the NRT product can be used to continue the LM-SIAge CDR after 2020. Nevertheless, it would be beneficial if OSI SAF extends the CDR product in the future and in the past (until 1979) to generate a continuous sea ice age product.

### 4.5 Validation results

Comparison of the LM-SIAge product with the ice drifting buoys shows (Figure 17, a and c) that in the majority of cases, the age of a buoy is lower than the collocated average or maximum ice age, which is as expected (see Section 3.9). In only 1.7 % of cases is the buoy age older than the maximum age of the LM-SIAge product. The map in Figure 17, b shows that the underestimation of max-age occurs most often at the ice edge, where the concentration and drift fields are the most uncertain, or on land-fast ice, where the buoys get stuck, but ice motion is detected on the coarse ice drift product. The NSIDC product does not underestimate the buoy age in 96.4% of cases (Fig. 17, d).

### 4.6 LM-SIAge data content

The dataset is provided in daily netCDF files, following the CF conventions (CF-community, 2022), on an EASE2 grid with a resolution of 25 km. The variables in the files are listed in Table 1.

## 5 Conclusions

We have produced a new Climate Data Record (CDR) of Arctic sea ice age, LM-SIAge, spanning the period from 1991 through 2024. MYI concentration data are available from 15 September 1991; and six sea ice age fractions are available after the spin-

**Table 1.** Variables in the netCDF files of the LM-SIAge dataset

| Name | Description | Dimensions | Precision |
|------|-------------|------------|-----------|
| x | X-coordinate of the projected dataset | x | double |
| y | Y-coordinate of the projected dataset | y | double |
| time | Reference time of product | time | double |
| sea_ice_age | Weighted Average of sea ice age | time,x,y | float |
| conc_1yi | Concentration of first-year ice | time,x,y | float |
| conc_2yi | Concentration of second-year ice | time,x,y | float |
| conc_3yi | Concentration of third-year ice | time,x,y | float |
| conc_4yi | Concentration of fourth-year ice | time,x,y | float |
| conc_5yi | Concentration of fifth-year ice | time,x,y | float |
| conc_6yi | Concentration of sixth-year ice | time,x,y | float |
| status_flag | Status flag for sea ice age. flag = 0: Nominal retrieval by the SIAge algorithm; flag = 1: Position is over land; flag = 2: Pixel is invalid. | time,x,y | byte |
| uncertainty_1yi | Total uncertainty of first-year ice | time,x,y | float |
| uncertainty_2yi | Total uncertainty of second-year ice | time,x,y | float |
| uncertainty_3yi | Total uncertainty of third-year ice | time,x,y | float |
| uncertainty_4yi | Total uncertainty of fourth-year ice | time,x,y | float |
| uncertainty_5yi | Total uncertainty of fifth-year ice | time,x,y | float |
| uncertainty_6yi | Total uncertainty of sixth-year ice | time,x,y | float |
| uncertainty_sia | Total uncertainty of sea ice age | time,x,y | float |

up period, starting from 15 September 1995. This dataset is derived using a novel Lagrangian advection scheme on a triangular mesh, allowing the tracking of multiple sea ice age fractions with minimal numerical diffusion. The product represents a significant advancement over previous efforts by providing a detailed, fraction-based view of sea ice age evolution at high spatial and temporal resolution. The dataset also provides detailed uncertainty estimates, with uncertainty increasing with ice age due to the cumulative effects of advection and drift inaccuracies.

Time series analysis shows consistent behaviour across other datasets. LM-SIAge reproduces major events such as the record-low MYI extent in 2007 and the near-disappearance of ice older than four years in 2012 (Regan et al., 2023). However, systematic differences remain: the NSIDC dataset has lower total ice area before 2011 and higher MYI area afterwards, while the C3S SIType CDR shows higher areas of both total and MYI ice, which at least partly is due to the area estimate accumulating entire pixels instead of fractional pixels.

Trend analysis from 1995 through 2024 reveals a consistent increase in first-year ice (FYI) area across all products, with LM-SIAge reporting a rate of 40,000 $\mathrm{km^2\,y^{-1}}$. In contrast, multi-year ice (MYI) area decreases by approximately -60,000 $\mathrm{km^2}$

$y^{-1}$ in both LM-SIAge and SIType CDR, while NSIDC underestimates this decline. Second-year ice trends are not statistically significant, and older ice types show a modest but persistent decline. Spatial trend analysis further reveals that the weighted average ice age has declined modestly, by 0.58 month $y^{-1}$. In contrast, maximum ice age—indicative of the oldest surviving ice—has decreased more sharply at rates of -1.43 to -1.92 month $y^{-1}$.

A comparison with ice drifting buoy data shows that the LM-SIAge product generally provides older age estimates than individual buoys, consistent with its area-weighted averaging approach. Discrepancies are most common near the ice edge, where observational uncertainties are highest.

In summary, the LM-SIAge dataset offers a robust, high-resolution record of Arctic sea ice age distributions and trends, with enhanced detail and reliability compared to existing datasets. It is suitable for climate monitoring, process studies, model evaluation, and assimilation into reanalysis systems. By providing a continuous and physically grounded record of sea ice age fractions, the dataset supports the growing need for detailed Arctic cryosphere indicators as part of the GCOS Essential Climate Variables framework. In future, it is planned to further improve the Lagrangian advection algorithm, include newer upstream CDRs, back-extend the time series up to 1979, and cover the Southern Ocean.

*Code and data availability.* The code for computing sea ice age using Lagrangian advection of a triangular mesh is publicly available on GitHub (Korosov and Edel, 2025b).

The monthly averaged LM-SIAge dataset version 2.1.2 described in this manuscript can be accessed at Zenodo under https://doi.org/10.5281/zenodo.15773500 (Korosov and Edel, 2025a).

The daily LM-SIAge dataset will be publicly available upon acceptance of the manuscript.

*Video supplement.* The first video supplement (Korosov, 2025a) provides a detailed demonstration of how advection on a triangular mesh is performed. Video "advect_disk.gif" shows the rotation of a disk with a diameter of 300 km and illustrates that diffusion occurs only at the edge of the disk. Video "advect_myi.gif" shows the advection of a MYI fraction from 15 September 1991, to 15 September 1992 on a pan-Arctic scale. Video "advect_myi_zoom.gif" displays the same data, but with a closer view of the Fram Strait.

The second video supplement (Korosov, 2025b) presents an animation of the entire sea ice age dataset, spanning from 15 September 1991, to 31 December 2024.

*Author contributions.* AK developed and ran the code for sea ice age computation. AK and LE developed and ran the data preprocessing code. EJD and TL provided early access to SID CDR drift data for experimentation. SA provided access to SIType data. All authors contributed to the writing of the manuscript.

*Competing interests.* We declare that none of the authors has any competing interests.

*Acknowledgements.* We are grateful for the support from the Research Council of Norway (project "TARDIS", no. 325241) and the European Space Agency (project "CCI SAGE", no. 4000147560/25/I-LR). We thank the EUMETSAT Ocean and Sea Ice Satellite Application Facilities, EU Copernicus Climate Change Service, and the U.S. National Snow and Ice Data Center for free access to the data. We acknowledge using Grammarly for correcting grammar and other mistakes.

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

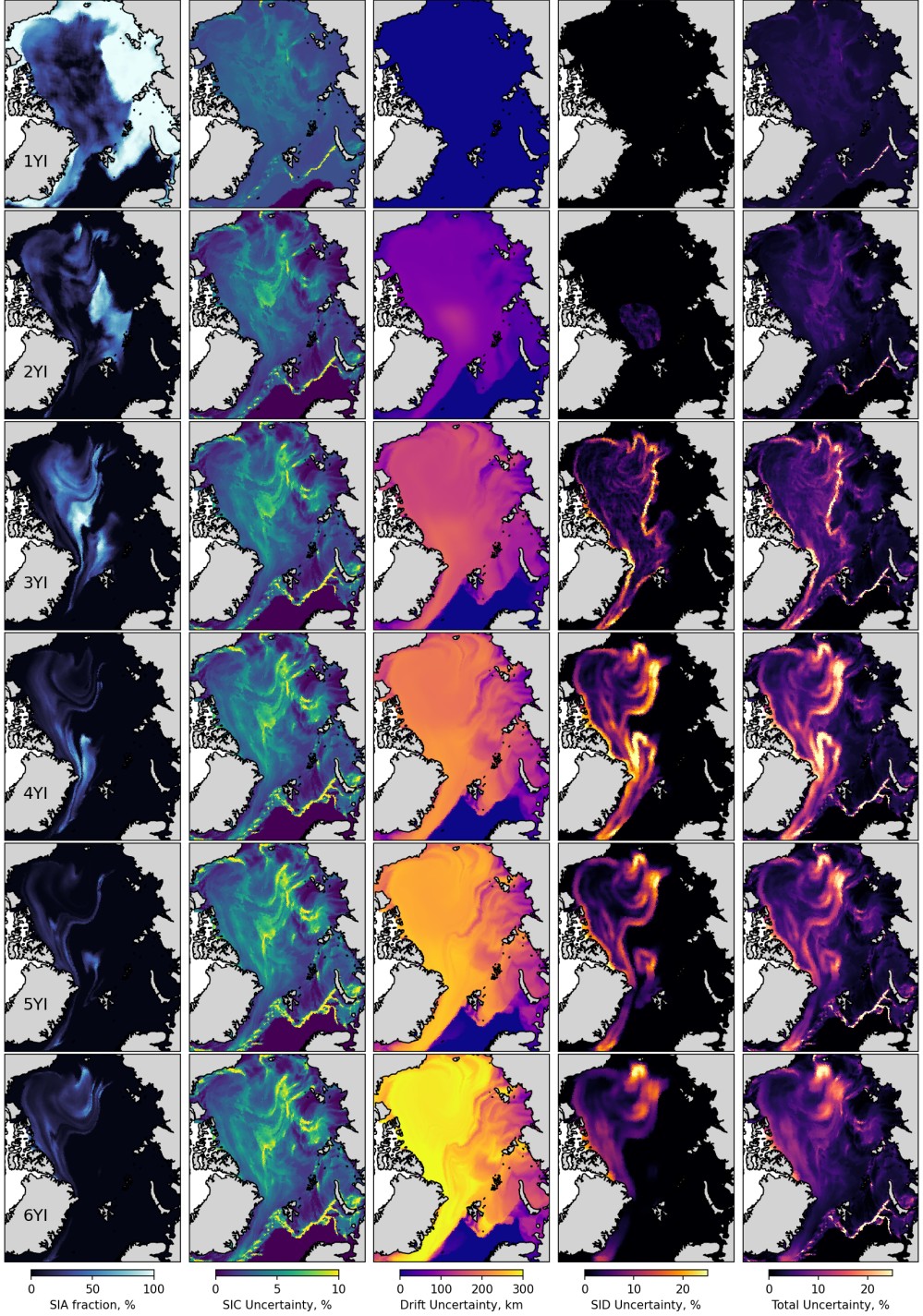

**Figure 9.** Maps of sea ice age fractions (first column), SIC uncertainties (second column), SID uncertainties (third column), uncertainties in SIC associated with SID (fourth column) and total uncertainties (fifth column) for 31 December 1995. Rows correspond to the first-year ice, second-year ice, etc. SIC uncertainties are provided as absolute values of concentration.

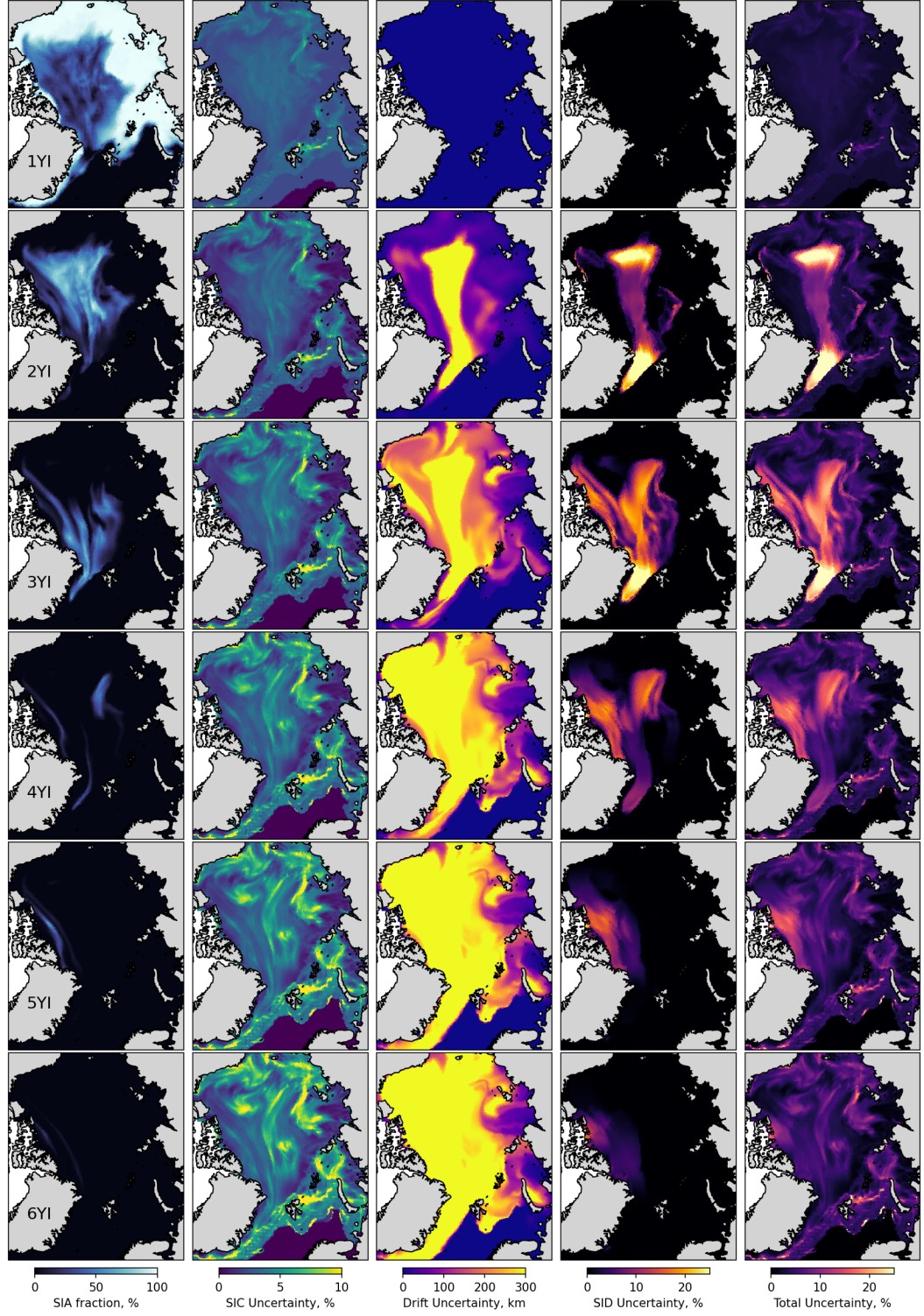

**Figure 10.** Same as Figure 9 but for 31 December 2024.

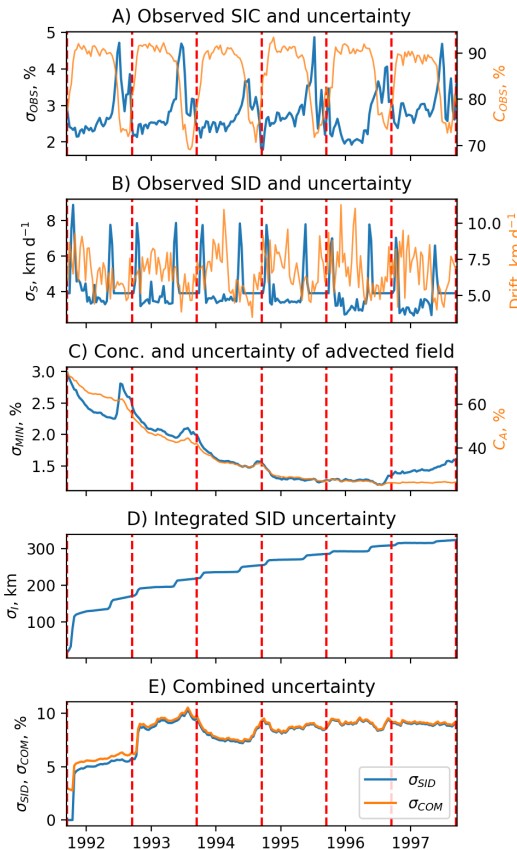

**Figure 11.** Variability of SIC and SID uncertainties in the MYI field advected from 15 September 1991 to 15 September 1997. A) Average observed concentration (orange, right axis) and its uncertainty (blue, left axis). B) Average observed drift speed (orange, right) and its uncertainty (blue, left axis). C) Average concentration of advected field (orange, left) and its uncertainty (blue, left axis). D) Integrated ice drift uncertainty. E) Uncertainty in concentration associated with ice drift uncertainty (blue) and total uncertainty (orange). Red dashed lines show 15 September.

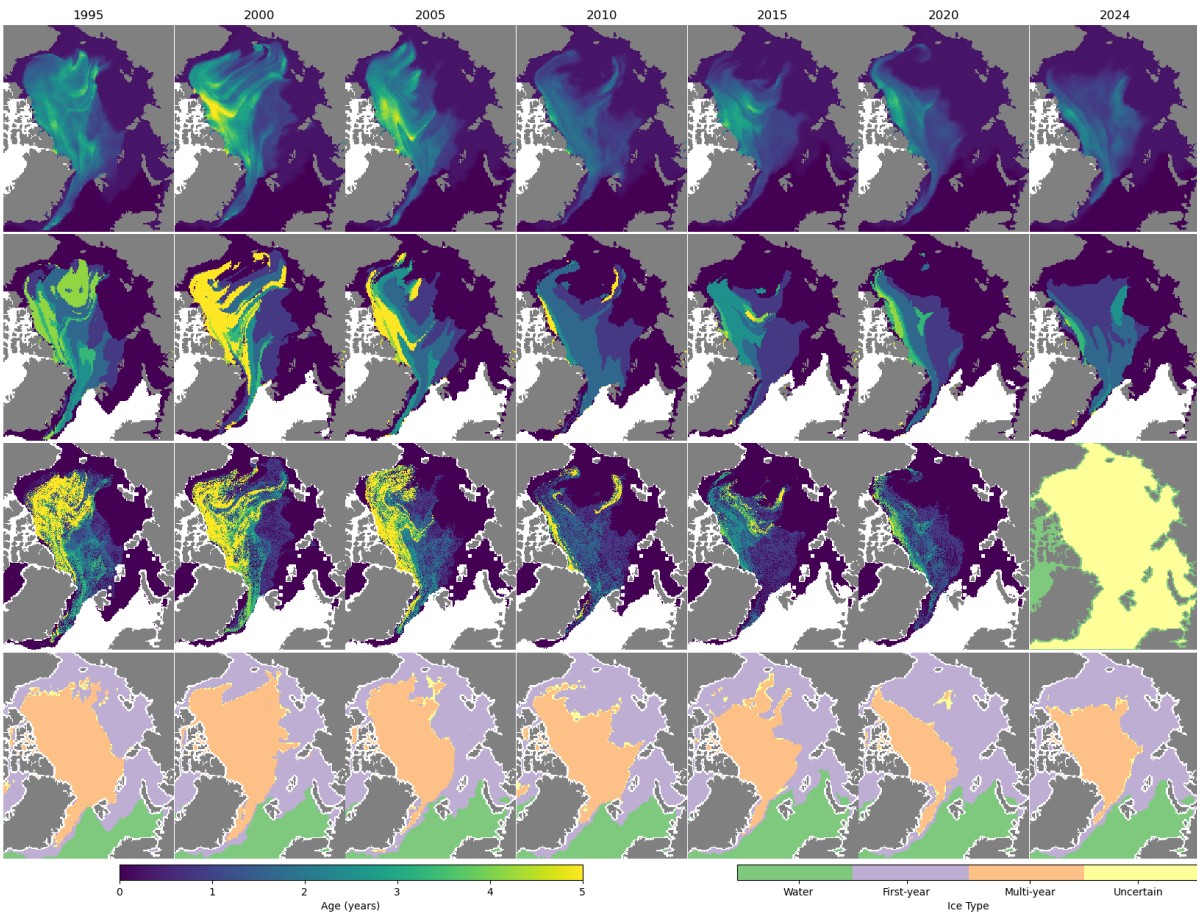

**Figure 12.** Maps of average ice age from the LM-SIAge (upper row), maximum ice age from LM-SIAge (second row), ice age from NSIDC (third row) and ice type from the SIType CDR (fourth row). Maps are provided for 31 December in the year shown in the upper row. The mask used for computing time series in Fig. 13 is shown in yellow on the third row.

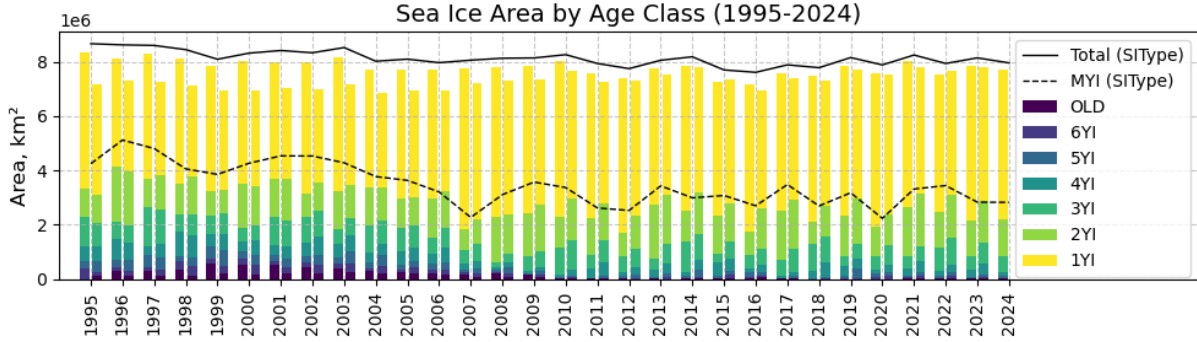

**Figure 13.** Time series of sea ice age area from LM-SIAge (bars on the left side for each year), NSIDC (bars on the right side for each year, except 2024) and SIType CDR (black lines). Values are given as of 31 December each year.

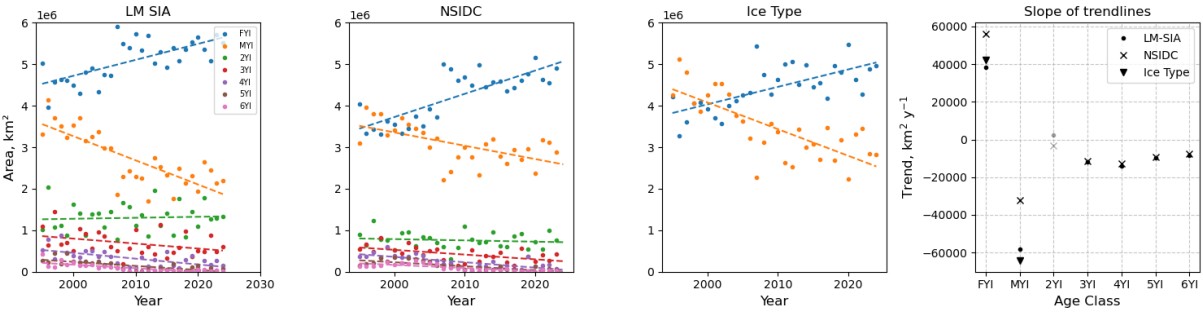

**Figure 14.** Trend analysis time series of ice age categories areas from LM-SIAge (left panel), NSIDC (second panel) and SIType CDR (third panel) datasets. The right panel displays the slopes of trend lines for each category (insignificant coefficients are shown in light grey).

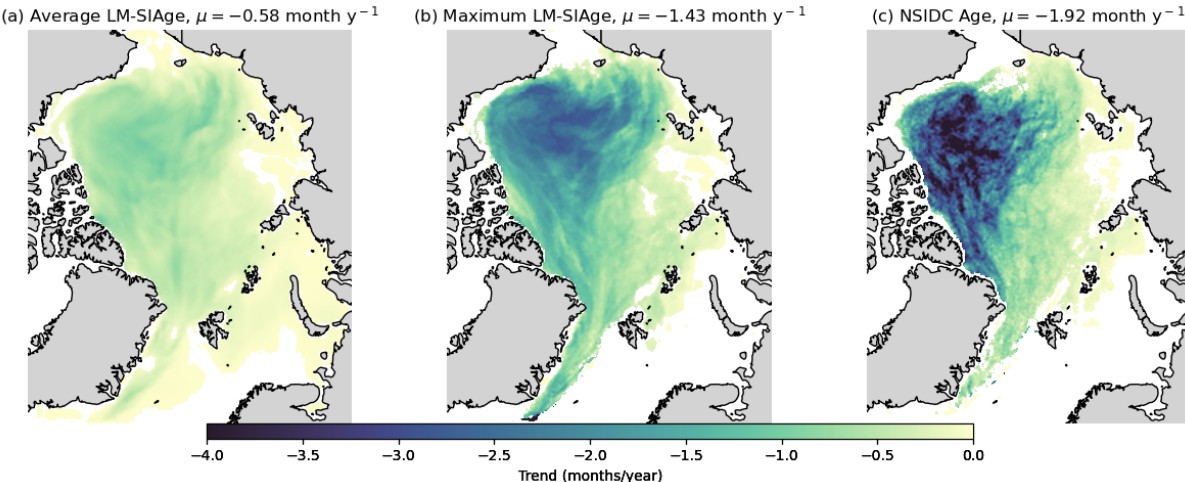

**Figure 15.** Linear trends in reduction of sea ice age [month/year] between 1995 – 2024 from three sources: weighted average LM-SIAge (a), max LM-SIAge (b), and NSIDC Age (b). Insignificant trends are masked by white colour.

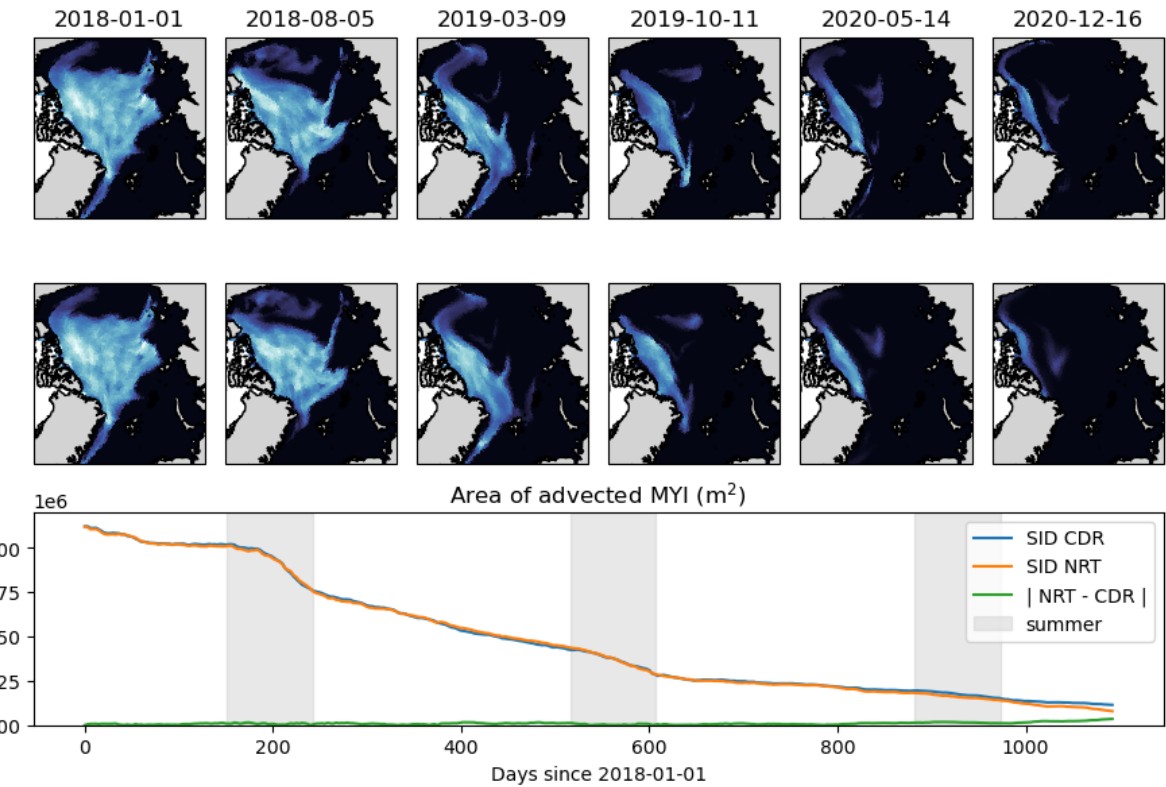

**Figure 16.** Comparison of MYI field from 2017 advected using SID CDR (upper row) and SID NRT product (second row). The lower row shows a comparison of areas of the advected MYI fields (km$^2$). The green line represents the absolute values of the area difference, and the grey shading indicates the summer period (June to August).

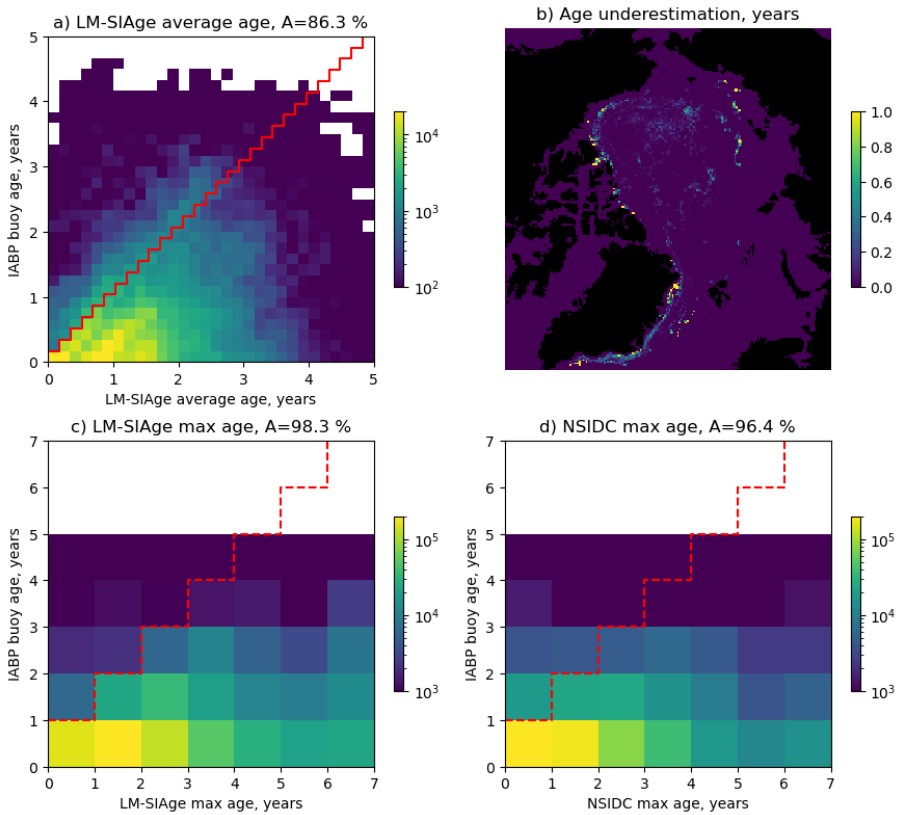

**Figure 17.** Comparison of LM-SIAge and NSIDC products with the age of the collocated ice drifting buoys. a) LM-SIAge weighted average age, c) LM-SIAge maximum age, d) NSIDC maximum age. The red line on a, c and d shows the separation between correct (below the line) and incorrect (above the line) LM-SIAge predictions. Accuracy ($A$) of each product is provided in the title. Colours denote the number of matchups. b) Map of the ice age is overestimation in the LM-SIAge dataset compared to the buoys, along with the magnitude of the discrepancy.