# Peer review of "A Climate Data Record of Sea Ice Age Using Lagrangian Advection of a Triangular Mesh"

_Earth System Science Data, 2025_

## Referee Comment (RC1)

Review of

A Climate Data Record of Sea Ice Age Using Lagrangian Advection of a Triangular Mesh

by Korosov, A, et al.

**Summary:**

The manuscript presents a new sea ice age data set based on a combination of PMW sea ice concentrations and sea ice drift products. This is an extension/substantial improvement of the earlier work where Eulerian approach to sea ice drift and fractional coverage of ice of different ages is now substituted by a Lagrangian sea ice advection scheme. The presented results on fractional ice ages for 1991-2024 are compared with other existing products from NSIDC and C3S. In general, this is a very exciting data product undoubtedly useful for a number of scientific and management applications.

The are a few comments that should be addressed for the manuscript to be published

Major to moderate comments.

1) Section 3.4 Mapping between the advected meshes

Eq. 5 and 6 suggest a fraction conservation during changes in the areal content (changes proportional to areal change). This can be the case during divergence, but during convergence and hence ice ridging this may not work. This scheme explicitly assumes the ridging intensity to be the same for different age classes. This is in general not the same as younger/thinner ice types have a higher chance for ridging during convergence.

The authors are entirely correct in their statement in Lines 185-186 that there is no straightforward way to account for this based on observations alone. It is not unlikely however that the discrepancies with other products can partly be a result of this equal scaling of fractions of different ages. This is a statement worth making in the text.

**2) Uncertainty calculations.**

Section 3.8 begins with presentation of "The uncertainty of the produced sea ice age variable". Please clarify what is "sea ice age variable". Is it something related with Eq.(11)? If this is the case, then (if I don't misunderstand something), Eq.12 does not seem to be correct - it is not just a sum of uncertainties scaled by the respective ages. Uncertainty of the weighted average can be calculated by the error propagation formula (can be found elsewhere).

Need to mention that this section is quite difficult to read/comprehend. Can the authors consider adding another figure similar to Figure 2 where the entire sequence of uncertainty calculation/aggregation is presented? As far as I see the authors assume the uncertainties for concentrations/fractions for different age classes to be independent?

The uncertainties, the way they are presented in the data (%) and in e.g. Figure 8, are these absolute values (i.e. ice concentrations) or % of these fractions/concentrations? What is the "total uncertainty" – sum of uncertainties for all ice fractions? Please clarify.

The authors apply caping to advected concentrations. Did the authors consider applying the same procedure (caping/or better say conditioning) to the uncertainties? The total uncertainty looks higher than the observed SIC uncertainty.

**Minor comments**

1) According to WMO nomenclature, see 2.6.1 in

https://cryo.met.no/sites/cryo/files/IceService\_docs/WMO\_259-2015\_multilingual.pdf

FYI ice that survived the summer minimum is called "residual ice" and it "officially" turns into SYI only on the 1 January of the coming winter. I understand that it makes it much easier for understanding if the indexing is changed in the way the authors did, but good to mention, at least, that you bypass the established classification a bit.

- 2) Figure 2: please add colorbars to the panels to improve the visualization. Consider also adding notation like "step 1", "step 2" (or just subplot numbers) in the figure and in the corresponding text in Lines 113->
- 3) Figure 15: please clarify what the numbers on the colorbars denote.

---

## Author Response (AR1)

**Authors' Response to Reviews of**

**A Climate Data Record of Sea Ice Age Using Lagrangian Advection of a Triangular Mesh**

Anton Korosov, Léo Edel, Heather Regan, Thomas Lavergne, Emily Jane Down, Signe Aaboe
*Earth System Science Data,* `ESSD-2025-477`
* * *
RC: *Reviewers' Comment*,    AR: Authors' Response,    ☐ Manuscript Text

We are grateful to the reviewers for a comprehensive review of our manuscript and detailed comments!

**1. Anonymous referee #1**

**1.1. Major to moderate comments**

RC: *(1) Section 3.4 Mapping between the advected meshes*
*Eq. 5 and 6 suggest a fraction conservation during changes in the areal content (changes proportional to areal change). This can be the case during divergence, but during convergence and hence ice ridging this may not work. This scheme explicitly assumes the ridging intensity to be the same for different age classes. This is in general not the same as younger/thinner ice types have a higher chance for ridging during convergence.*
*The authors are entirely correct in their statement in Lines 185-186 that there is no straightforward way to account for this based on observations alone. It is not unlikely however that the discrepancies with other products can partly be a result of this equal scaling of fractions of different ages. This is a statement worth making in the text.*

AR: Yes, we agree that not accounting for different convergence rates for different ice age fractions can be one of the reasons for the discrepancies. The following text is added in Results and discussions, at the end of a new sub-section **4.2 Comparison of LM-SIAge, NSIDC and SIType datasets**

> Several factors lead to the discrepancies observed between the LM-SIAge, NSIDC and SIType products. Firstly, the LM-SIAge and NSIDC are derived from MYI advection, whereas SIType is a radiometric product. Next, LM-SIAge and NSIDC use quite different ice drift products, advection schemes, and representations of the ice age state. In addition, LM-SIAge provides the MYI concentration for each pixel. Summing the areal coverage of MYI, weighted by its concentration, gives the total MYI area. In contrast, NSIDC and SIType products provide a categorical classification, assigning a fixed ice-age class to each pixel. In these cases, the total MYI coverage is obtained by summing the areas of all pixels classified as MYI, which corresponds more closely to an MYI extent. As a result, the ice extent is generally larger than the ice area.
> Finally, we don't account for different convergence (and melting) rates of ice of different ages, as we cannot constrain that by observations (see Eq. 5). Assuming that older ice is thicker and can converge (melt) less than the thinner younger fractions, we may overestimate the loss of older ice in converging (melting) cells. Our previous experiments with a numerical model-based estimate of sea ice age (Regan et al., 2023) indicate that ridging younger ice first yields more realistic estimates of MYI extent. However, that may lead to an underestimation of MYI ridging in areas where MYI and

> FYI thicknesses are similar (e.g., the marginal ice zone) and in recent years, as MYI thins out faster than FYI (Kwok et al., 20018).

**RC:** *(2) Uncertainty calculations.*
*Section 3.8 begins with presentation of "The uncertainty of the produced sea ice age variable". Please clarify what is "sea ice age variable" . Is it something related with Eq.(11) ? If this is the case, then (if I don't misunderstand something), Eq.12 does not seem to be correct - it is not just a sum of uncertainties scaled by the respective ages. Uncertainty of the weighted average can be calculated by the error propagation formula (can be found elsewhere).*

**AR:** Yes, indeed, we did not account for division by the sum of concentrations. The error propagation formula in the case of division/multiplication reads as follows:

$$\left(\frac{\sigma_q}{q}\right)^2 = \left(\frac{\sigma_x}{x}\right)^2 + \left(\frac{\sigma_y}{y}\right)^2, \tag{1}$$

where $q = x/y$. I.e., $x = \sum_{i=0}^{N} A_i C_{iY}$, $y = \sum_{i=0}^{N} C_{iY}$, $\sigma_x^2 = \sum_{i=0}^{N} A_i * \sigma_{C_{iY}}^2$, $\sigma_y^2 = \sum_{i=0}^{N} \sigma_{C_{iY}}^2$.
The equation is corrected in the text, and the uncertainties are recalculated in the product.

**RC:** *Need to mention that this section is quite difficult to read/comprehend. Can the authors consider adding another figure similar to Figure 2 where the entire sequence of uncertainty calculation/aggregation is presented? As far as I see the authors assume the uncertainties for concentrations/fractions for different age classes to be independent?*

**AR:** The uncertainty computation section is rewritten from deductive (from general principles to specific conclusions) into inductive (from specific details to general conclusions) logic for easier understanding. A flowchart with an uncertainty estimation algorithm is added (see Fig. 1 below). Yes, we assume the uncertainties are independent because we don't have a method to compute their covariations. This is indicated at the beginning of the section.

**RC:** *The uncertainties, the way they are presented in the data (%) and in e.g. Figure 8, are these absolute values (i.e. ice concentrations) or % of these fractions/concentrations? What is the "total uncertainty" – sum of uncertainties for all ice fractions? Please clarify.*

**AR:** These are absolute values, concentrations. The following clarification is added to the figure caption:

> SIC uncertainties are provided as absolute values of concentration.

**RC:** *The authors apply caping to advected concentrations. Did the authors consider applying the same procedure (caping/or better say conditioning) to the uncertainties? The total uncertainty looks higher than the observed SIC uncertainty.*

**AR:** In case of conditioning, we keep the minimal value of the two concentrations. We therefore have to take the full uncertainty of the value we keep, not the minimum of the two uncertainties. The total uncertainty (i.e., left column in Figs. 8 and 9) is a combination of SIC uncertainty (the second column) and SID uncertainty (fourth column), and is therefore larger than both of them (See Eq. 15). As the uncertainty computation section is completely rewritten, it is easier to understand the relation between the uncertainties presented in Figs. 8 and 9.

[Figure]

Figure 1: Flowchart of computing the uncertainty of sea ice age. The input data is shown in yellow, and the final result is displayed in green. The orange arrows indicate data flow using the advected mesh. See Eqs. 12 - 19 for the notation of individual uncertainty components.

**1.2. Minor comments**

**RC:** *(1) According to WMO nomenclature, see 2.6.1 in* `https://cryo.met.no/sites/cryo/files/IceService_docs/WMO_259-2015_multilingual.pdf` *FYI ice that survived the summer minimum is called "residual ice" and it "officially" turns into SYI only on the 1 January of the coming winter. I understand that it makes it much easier for understanding if the indexing is changed in the way the authors did, but good to mention, at least, that you bypass the established classification a bit.*

AR: The following text is added in Section 3.1, after Eq. 2:

> It should be noted that according to the nomenclature of the World Meteorological Organisation (Sea Ice Nomenclature, WMO-259), the first-year ice (FYI) that survives the summer minimum is called "residual ice" and it turns into second-year only on 1 January of the coming winter. Nevertheless, hereafter, all survived FYI turns into second-year after 15 September.

**RC:** *(2) Figure 2: please add colorbars to the panels to improve the visualization. Consider also adding notation like "step 1", "step 2" (or just subplot numbers) in the figure and in the corresponding text in Lines 113->*

AR: Two colobars (for the advected and computed concentrations) are added to the figure. Notation of steps is added to the figure and is referenced in the accompanying text.

**RC:** *(3) Figure 15: please clarify what the numbers on the colorbars denote.*

AR: The following clarification is added to the figure caption:

SIC uncertainties are provided as absolute values of concentration.

**2. Anonymous referee #2**

**RC:** *(1) One thing that I was concerned about was a validation and uncertainty analysis of the constructed dataset. The authors used buoy position data to validate the age product. It is an acceptable approach. However, the used buoy trajectories cover only between 2002 and 2024. The product for the earlier period of 1991 – 2011 was not validated in this manuscript. The uncertainty inherent in this product depends on the quality of the passive microwave-based SICs. The uncertainty of SICs can vary with seasons. The seasonal uncertainties of the age product due to the SIC uncertainties should be addressed. In addition, the uncertainty section is difficult to read and understand. I recommend reformulating this section.*

**AR:** The buoy database is extended to start from 1991, and the validation results are updated. The uncertainty computation section is rewritten from deductive (from general principles to specific conclusions) into inductive (from specific details to general conclusions) logic for easier understanding. A flowchart with an uncertainty estimation algorithm is added (see Fig. 1). Analysis of seasonal variations of uncertainty is added as a sub-section in Results:

> **4.3 Seasonal and interannual variations of uncertainty**
> We analysed the variability of average uncertainty in the source data and in the derived dataset (see Fig. 2). The observed SIC uncertainty ($\sigma_{C_{Obs}}$, Fig. 2, A) shows strong seasonal variations, with a minimum ($\approx 2\%$) in winter and a maximum ($\approx 4.5\%$) during the melt season. The observed SID uncertainty ($\sigma_S$, Fig. 2, B) also has a minimum ($\approx 3.5$ km d$^{-1}$) in winter, a plateau of constant values of 4 km d$^{-1}$ in summer, and two peaks with $\approx 7$ km d$^{-1}$ just before and after the summer period. The uncertainty of the advected MYI field ($\sigma_{SIC}$, Fig. 2, C) starts from $\approx 3\%$ and gradually decreases over 6 years, with a slight increase during summer seasons. The integrated uncertainty of ice drift ($\sigma_I$, Fig. 2, D) starts from nearly zero and increases step-wise following the pre- and post-summer peaks of $\sigma_S$. The uncertainty of advected MYI concentration, associated with the uncertainty of ice drift ($\sigma_{SID}$, Fig. 2, E), also begins low and then rapidly increases during the first year, which is followed by a gradual increase during consecutive years with strong seasonal variations. The total uncertainty of the advected MYI field ($\sigma_{C_A}$, Fig. 2, E) is first dominated by the uncertainty in the observed SIC, but after the end of the melt season and the jump of $\sigma_S$, the contribution of $\sigma_{SIC}$ becomes much less pronounced.

**RC:** *(2) In methodology, the authors chose certain parameters, e.g., mesh element size thresholds, angle, and element area, which seem to be used without any justification. It would be better if the authors provide how sensitive the results are changed due to these parameters or references to support these values.*

**AR:** Values for these parameters are chosen to keep the size of the mesh elements similar to the resolution of the destination grid (i.e., $\Delta x = \Delta y =25$ km). Smaller parameter values yield smaller elements and do not change the results, but they require much more CPU time for advection and, especially, remeshing. Larger values/elements lead to a lower effective resolution of the product (neighbour pixels on the destination grid have the same values. The following clarification is added after the parameter values are listed:

> The end results are not very sensitive to the mesh size, and these parameter values are chosen to keep the area of the mesh elements below the area of the destination grid elements with a spatial

[Figure]

Figure 2: SIC and SID uncertainties.

resolution of 25 km, and to keep the elements large enough for efficient advection and, especially, time-consuming remeshing.

**RC:** *(3) Are there any possibilities of ridging or deformation processes with different age categories affecting the proposed algorithm?*

**AR:** Yes, not accounting for different convergence rates for different ice age fractions can be one of the reasons for the discrepancies. The following text is added in Results and discussions, at the end of a new Sub-section **4.2 Comparison of LM-SIAge, NSIDC and SIType datasets**

Several factors lead to the discrepancies observed between the LM-SIAge, NSIDC and SIType products. Firstly, the LM-SIAge and NSIDC are derived from MYI advection, whereas SIType is a radiometric product. Next, LM-SIAge and NSIDC use quite different ice drift products, advection schemes, and representations of the ice age state. In addition, LM-SIAge provides the MYI concentration for each pixel. Summing the areal coverage of MYI, weighted by its concentration, gives the total MYI area. In contrast, NSIDC and SIType products provide a categorical classification, assigning a fixed ice-age class to each pixel. In these cases, the total MYI coverage is obtained by summing the areas of all pixels classified as MYI, which corresponds more closely to an MYI extent. As a result, the ice extent is generally larger than the ice area.
Finally, we don't account for different convergence (and melting) rates of ice of different ages, as we cannot constrain that by observations (see Eq. 5). Assuming that older ice is thicker and can converge (melt) less than the thinner younger fractions, we may overestimate the loss of older ice in converging (melting) cells. Our previous experiments with a numerical model-based estimate of sea ice age (Regan et al., 2023) indicate that ridging younger ice first yields more realistic estimates of MYI extent. However, that may lead to an underestimation of MYI ridging in areas where MYI and FYI thicknesses are similar (e.g., the marginal ice zone) and in recent years, as MYI thins out faster than FYI (Kwok et al., 2018).

**RC:** *(4) It is very fair to provide Figure 15 for the NSIDC product as well.*

**AR:** The NSIDC age product is validated against the IABP data using the same methodology as for the LM-SIAge product (see Fig. 3 below). There are fewer correct predictions of the maximum ice age in the NSIDC product than in the LM-SIAge. Moreover, collocations of IABP buoys younger than 1 year with corresponding NSIDC estimates are almost absent, suggesting a positive bias in the NSIDC product. The following text is added in the abstract:

Validation with ice drifting buoys indicates good consistency (LM-SIAge does not underestimate max age of the buoys in 98.3%), with most discrepancies occurring near the ice edge. The NSIDC product does not underestimate the age only in 96.4% and potentially underestimates the presence of FYI.

The following text is changed in the section **3.9 Validation**:

For validation of the LM-SIAge  and NSIDC products, we used trajectories of sea ice drifting buoys from the IABP dataset. We compare the maximum ice age detected by  the Lagrangian algorithms to the age of the ice drifting buoys.

The following text is added in the section **4.3 Validation results**:

[Figure]

Figure 3: Validation of the Lagrangian ice age products on ice drifting buoy data.

The NSIDC product does not underestimate the buoy age in 96.4% (see Fig. 15, D). Comparing 2D histograms on Fig. 15 B and D, we observe a significantly lower number of match-ups in the NSIDC FYI category, confirming that this product tracks the maximum age in a grid cell and potentially underestimates the presence of FYI.

**RC:** *(5) In supplementary videos, I found that somewhat permanent multiyear ice exists in the coastline of the Kara Sea and Islands between the Kara and Laptev Seas. Is this correct?*

AR: Thank you for spotting this! MYI is not expected to appear that far south. However, this is not an error in the Lagrangian advection algorithm. That happens because non-zero sea ice concentrations are present in the upstream SIC CDR product year-round due to, most likely, the spill-over effect. We added a flag to the output product indicating which pixels are likely affected by this effect and may have falsely high MYI concentrations. The following text was added to sub-section **4.2 Comparison of LM-SIAge, NSIDC and SIType datasets**:

One minor difference between the LM-SIAge and NSIDC products, which is hard to spot, is the presence of MYI near the coast in the Kara and Laptev seas (also seen on the supplementary videos). This is not realistic and results from enhanced sea ice concentrations near the coast in the upstream SIC products due to the "land spillover" effect (Kern et al., 2022). These pixels are masked in the netCDF files.

**3. Anonymous referee #3**

**3.1. Specific comments**

**RC:** *Section 3.1 – Representation of Age Evolution*
*The computation of sea-ice age change appears to assume that all ice categories evolve similarly in time. In reality, when sea-ice concentration decreases, younger ice tends to melt more rapidly than older ice. It would be helpful if the authors could clarify whether this differential melting behavior is accounted for in their formulation, or discuss its potential implications for the resulting age distribution.*

**AR:** Indeed, not accounting for different convergence and melting rates for different ice age fractions can add a bias to the age distributions. The following text is added in Results and discussions, at the end of a new Sub-section **4.2 Comparison of LM-SIAge, NSIDC and SIType datasets**

> Several factors lead to the discrepancies observed between the LM-SIAge, NSIDC and SIType products. Firstly, the LM-SIAge and NSIDC are derived from MYI advection, whereas SIType is a radiometric product. Next, LM-SIAge and NSIDC use quite different ice drift products, advection schemes, and representations of the ice age state. In addition, LM-SIAge provides the MYI concentration for each pixel. Summing the areal coverage of MYI, weighted by its concentration, gives the total MYI area. In contrast, NSIDC and SIType products provide a categorical classification, assigning a fixed ice-age class to each pixel. In these cases, the total MYI coverage is obtained by summing the areas of all pixels classified as MYI, which corresponds more closely to an MYI extent. As a result, the ice extent is generally larger than the ice area.
>
> Finally, we don't account for different convergence (and melting) rates of ice of different ages, as we cannot constrain that by observations (see Eq. 5). Assuming that older ice is thicker and can converge (melt) less than the thinner younger fractions, we may overestimate the loss of older ice in converging (melting) cells. Our previous experiments with a numerical model-based estimate of sea ice age (Regan et al., 2023) indicate that ridging younger ice first yields more realistic estimates of MYI extent. However, that may lead to an underestimation of MYI ridging in areas where MYI and FYI thicknesses are similar (e.g., the marginal ice zone) and in recent years, as MYI thins out faster than FYI (Kwok et al., 20018).

**RC:** *Clarity of Method Description (Figures 5 and 6)*
*Since the triangular-mesh approach and its associated remeshing procedure may be unfamiliar to many readers, the description of these processes could be improved for clarity. Figures 5 and 6 are central to understanding the proposed algorithm, but both could be made more legible and intuitive.*

**AR:** Figure 5 and 6 were updated according to the specific comments below (see Figs. 4 and 5). References to specific panels on figures 5 and 6 were added to the text for clarity.

**RC:** *In Figure 5, particularly in the left and right examples, it is difficult to visually identify what has changed before and after the remeshing process. The green lines representing the remeshed state are also hard to distinguish.*

**AR:** Descriptive labels were added to Figure 5 for easier understanding, and the green colour was changed to yellow (see Fig. 4 below).

**RC:** *In Figure 6, the blue and black lines in the left panel are not easily distinguishable. Enhancing the color contrast or line thickness would improve readability.*

[Figure]

Figure 4: Scheme of three types of remeshing: collapsing of a short edge (A), splitting of a long edge (B), removing a flipped element (C). Edges and elements before remeshing are shown in red in the upper row, and the new mesh is shown in yellow in the lower row.

AR: Three panels in Figure 6 were replaced with four panels, illustrating the advection/remeshing/optimisation processes in more detail and using only two colours for clarity (see Fig. 5 below).

**RC:** *Adding brief explanatory captions to Figures 5 and 6 that summarize what each step represents (e.g., "before remeshing," "after remeshing," "regularized mesh") would help non-specialist readers.*

AR: Clear indication of remeshing process steps is added to the figures and their captions (see Figs. 4 and 5 below).

**4. Handling topic editor, Clare Eayrs**

**4.1. Uncertainty quantification and presentation**

**RC:** *All reviewers found the section on uncertainty difficult to follow. Please reformulate this section and consider adding a flowchart or schematic that illustrates the complete sequence of uncertainty estimation. Please define exactly what is meant by "sea ice age variable" and explain how uncertainties are calculated and combined.*

AR: The uncertainty computation section is rewritten from deductive (from general principles to specific conclusions) into inductive (from specific details to general conclusions) logic for easier understanding. A flowchart with an uncertainty estimation algorithm is added (see Fig. 1). Only a flowchart is provided in the current response, and the rewritten section will be submitted with the revised manuscript.

[Figure]

Figure 5: Illustration of the node advection and mesh optimisation process. A) Some nodes of the initial mesh (shown in black) are advected using ice drift vectors (shown in red). B) The advected mesh (shown in black) has some distorted elements that require remeshing (shown in red). C) Most of the elements on the remeshed mesh remain unchanged (shown in black). The new elements introduced by remeshing are shown in red. D) Position of the nodes in the remeshed mesh is updated, and a regularised mesh is created (shown in black). The previous mesh (remeshed, but not regularised, shown in red) differs from the optimised one only near the new elements, in the vicinity of the convergence/divergence zone. In contrast, in the homogeneous ice-drift area (lower right corner), the advected mesh is equal to the remeshed and optimised meshes.

**RC:** *If the uncertainty varies spatially and temporally, please provide the uncertainty as a separate data file that matches the spatial and temporal dimensions of your primary product, and document how users should apply the uncertainty information.*

**AR:** The spatially and temporally varying uncertainty is already provided together with the product. For convenience, it is included in the data files, as specified in Table 1 of the submitted manuscript. In addition, we suggest adding a subsection to Results and discussion that highlights seasonal and interannual variations in uncertainty (see Fig. 2 and our response to refree #3 above).

**RC:** *Please clarify whether the uncertainties shown in figures are absolute or relative values, and whether the "total uncertainty" represents the sum or the propagated metric. Please also address Reviewer 1's query regarding capping/conditioning of the uncertainty.*

**AR:** The uncertainties are absolute, and a corresponding notation is added to the figure caption. We addressed the Reviewer 1's query regarding capping and improved explanations in the rewritten section on uncertainty computations.

**4.2. Physical assumptions in the algorithm**

**RC:** *Reviewers 1 and 2 both highlighted that your scheme assumes equal scaling of fractions of different age classes. Please discuss the limitations of the mapping approach during convergence/ridging and how the assumption of equal scaling fractions of different age classes could influence the product and comparisons with other datasets. Reviewer 3 also asked whether your formulation accounts for the fact that younger ice tends to melt more rapidly when the concentration decreases. Please clarify whether this differential melting is represented, and if not, what the likely effect is on the age distribution.*

**AR:** The clarification on how equal convergence and melting of ice of different ages affect the final results has been added.

**4.3. Justification for parameter choices**

**RC:** *Please include a justification for the numerical parameters in the mesh set-up and either add references or a brief discussion on the sensitivity of these parameters to support these values.*

**AR:** The parameter values are chosen using simple geometric and algorithm efficiency considerations. Their values do not significantly affect the results, but may increase processing time or reduce the effective spatial resolution. Corresponding clarifications are added to the text.

**4.4. Figures**

**RC:** *Please add labels to colorbars and address the specific review comments.*

**AR:** The labels and colobars are added as requested. Not all the updated figures are presented in the current response.

**RC:** *Please specify the DOIs for the OSISAF datasets that the user should download in the README file in the Zenodo repository. I noticed you recommend the Sea Ice Concentration Climate Data Record Release 3 (OSI-450-a), but this has been superseded by version 3.1 (OSI-450-a1). I strongly encourage the authors to update the sea ice age product to base it on the latest sea ice concentration CDR, thereby making it more useful for downstream users. An update now would also help prevent confusion with future work based on newer data and likely increase the dataset's usage and citations. If this update is not feasible at this stage,*

*it would be helpful to outline any plans for future updates.*

AR: The DOIs of the upstream data products are added to the Zenodo README. The current dataset was produced before SIC CDR v3.1 was available. Unfortunately, it is not feasible to switch to a new version because downloading and reprocessing would require resources we do not currently have. We are running a project that provides funding to improve algorithms and extend the dataset to include data through 1978. Understandingly, this process will take some time. We, therefore, would like to publish the dataset as is, with a minor update to flag spurious MYI. The following text is added in Conclusions regarding our plans for future updates:

> In future, it is planned to improve the Lagrangian advection algorithm, include newer upstream CDRs, back-extend the time series up to 1979, and cover the Southern Ocean.

**5. Signe Aaboe**

RC: *I continued to read throughout the paper and have a few comments to share while still fresh in my mind. First of all, this is a great paper and exciting work. Congratulations on the great LM-AGE algorithm and the long CDR! It will be interesting to see what new age-analyses for the Arctic will show with these more detailed data, including uncertainty estimates.*

AR: Thank you :)

RC: *I wanted to share some thoughts about Figure 11. Did you try to calculate the sea-ice-age extent for LM-AGE? As I read the text and how I understand the area calculations, there is a difference from LM-AGE, which takes sea-ice concentrations into account in the computation, versus NSIDC and SIType, which both have only one class for the entire pixel, independent of the sea-ice concentration in that pixel. E.g. a pixel will show 100% multiyear ice in a pixel even though the ice concentration may be only 50%. So, the numbers for NSIDC and SIType are, in fact, sea-ice age extents compared with LM-AGE sea-ice age area. Extents will typically be larger than Area. This may explain why SIType overestimates LM-AGE.*

AR: I agree. We use ice area (sum of concentrations), while SIType and NSIDC use ice extent (sum of pixels). This is reflected now in Section 4.3 Comparison of LM-SIAge, NSIDC and SIType dataset.

RC: *In lines 7-10 on page 11, it says that the same mask is used. Would it be an idea to show the mask used? For instance, there are apparent differences in the areas covered by the various products across the Canadian Archipelago and Baffin Bay. Also, the coverage towards land differs and could potentially explain some of the total area differences between LM-AGE and NSIDC?*

AR: I added the mask to Fig. 12.

RC: *Finally, after reviewing the results, it seems that the recent version 4 sea-ice type CDR was used in the paper, which is really nice to see. Thanks. I made a few edits to the description and updated the references:*

The sea-ice type CDR (version 4; Aaboe et al., 2023a), downloaded from the C3S Climate Data Store (Aaboe et al., 2023b), is a daily classification product that maps the dominant ice types, first-year ice, multiyear ice, and an ambiguous ice class, across the Arctic at 25 km resolution. Here, multiyear ice is defined as all ice that has survived at least one summer melt, corresponding to a sea-ice age of 2 years or more. It is derived using a Bayesian classification algorithm applied to passive microwave brightness temperatures from the SMMR, SSM/I, and SSMIS (CM SAF FCDR), combined with atmospheric reanalysis data (ERA5) and auxiliary sea-ice information. The product applies a temperature-based correction scheme to reduce misclassification

of ice types in situations where warm air intrudes into sea-ice-covered regions. It also incorporates sea-ice drift information both to correct misclassifications through a backtracking scheme and to refine the daily tuning of the algorithm. The ice-type classification is available only during the winter months (October-April) and covers the period from 1978 to the present. The product is provided on an EASE2 25 km grid.

References

AR: I updated the description.

RC: *Figure2. There are missing some arrows for the three rightmost subplots. Also, it is not obvious what the colors represent. Consider adding the two types of colorbar somewhere in the plot to show that white and yellow are the highest values/highest concentration of the class. Or at a minimum, describe this in the figure text.*

AR: I added the arrows and the colorbars.

RC: *Missing a first-time definition of FYI and also MYI before being used in e.g. L106? The way these terms are used in the manuscript, I think it is important to define them early as "ice that has not passed a summer season" and "ice that has survived at least one summer melt" (which is in fact different than e.g. WMO nomenclature)*

AR: I added definition of MYI and FYI and mentioned the difference from WMO.

RC: *Regarding our recent discussions in SAGE would you consider rewording in e.g. L108 to something like "which has survived at least two summer melt seasons". And should "C_(1Y)" then be labelled "C_0S" as for the 0 year/summer and no d used. And again in L111 "second-year ice (ice that has survived only one summer melt)". This is a more precise definition and could be labelled "C_1S" to follow our SAGE discussions.*

AR: I use the definition of the first-year ice as formed during the ongoing season and the second-year ice as surviving one melting season. I added an explicit definition to the text. In this manuscript, I prefer to use indices $_{1Y}$ for the first year and $_{2Y}$ for the second year (also explicitly indicated in the text now) to avoid confusion. Our SAGE discussions are still ongoing, and I believe that labelling the first-year as $C_{0S}$ is misleading.

RC: *I just saw figure 3, which is new(?) and is good since it also includes an extra time step to show the full advection process. However, I would suggest labeling them "C_1S" for the second-year ice and "C_2S" for the third-year ice (following the SAGE discussion on how many summer seasons it has survived not to be misunderstood as how many years the ice is).*

AR: As I mentioned above I'd rather keep $_{1Y}$ and $_{2Y}$.

RC: *The description from L113 uses different variables than shown in the figure, C_MY versus MYI. C_MY makes more sense and I suggest to implement this in the figure as well.*

AR: I agree, the figure is updated.

RC: *eq4 - are these subscript labels correct?*

AR: Nice spot! Indices updated.

RC: *I might have missed it in the text, but could you explain how, for example, second-year ice develops throughout the winter season? Is it advected by drift, or is it calculated in each time step, as the subtraction*

***between two advected MYI fields? (I figure 3 the green boxes do not show any evolution)***

AR: We do not advect the second-year ice itself. We rather advect the MYI fields from the previuos years ($C_{A1}$) and from the one before the previous ($C_{A2}$), etc, then the concentration of the second year ice is computed at each tep using the observed (total) and the advected concentrations $C_{2Y} = C_{OBS} - C_{A1}$.

---

## Author Response (AR2)

**Authors' Response to Reviews of**

**A Climate Data Record of Sea Ice Age Using Lagrangian Advection of a Triangular Mesh**

Anton Korosov, Léo Edel, Heather Regan, Thomas Lavergne, Emily Jane Down, Signe Aaboe
*Earth System Science Data,* ESSD-2025-477
* * *
**RC:** *Reviewers' Comment*,    AR: Authors' Response,    □ Manuscript Text

We are grateful to the reviewers and the editor for a comprehensive review of our manuscript and detailed comments!

**1. Comments from the anonymous referee and the editor**

**RC:** *1. The reviewer notes that while the revised uncertainty description and formulas are much clearer and the error propagation approach is correct and well explained, Equation 12 could benefit from additional clarification. The indices may be confusing for some readers. Please revise this equation or add explanatory text to ensure the notation is unambiguous.*

**AR:** The notation of the processing step was changed according to the previous notation in Eqs. 7 and 8, and a supporting comment is added.

$$\sigma_n^2 = k_n \sigma_{n-1}^2 \quad \sigma_{n+1}^2 = k_{n+1} \sigma_n^2$$

**RC:** *2. The new text explains the Lagrangian scheme limitations. However, the reviewer suggests adding one short paragraph that clearly states the main limitations for users (e.g coastal areas, early-1990s drift quality and strong deformation zones).*

**AR:** The following paragraph was added to **Conclusions**

> Among the limitations of the LM-SIAge dataset, we can name the following. First, although MYI concentration has been available since 15 September 1991, the full set of ice age fractions is available only after the spin-up period, starting from 15 September 1995. Second, the different convergence (and melting) rates of ice of various ages are not accounted for due to the absence of systematic observations for constraining these rates. Assumption of similar convergence (melt) rate for older and younger ice may lead to overestimation of the loss of older ice. And finally, the unrealistic presence of MYI near the coast in the Kara and Laptev Seas results from enhanced sea ice concentrations near the coast in the upstream SIC products due to the "land spillover" effect.

**RC:** *3. Please add a brief mention that buoy coverage in the 1990s is limited, and therefore validation in this earlier period is less robust than after 2002. This will help users appropriately interpret the validation results across the full temporal range.*

**AR:** The following text is added to the sub-section **2.5 Ice drifting buoy trajectories**

> As noted in (Lavergne and Down, 2023), the overall coverage of the Arctic Ocean with buoys throughout the 3 decades is good, with fewer observations in the peripheral Arctic seas. However, the number of buoys before 2002 is somewhat smaller, making validation in this earlier period less robust.

**RC:** *4. Finally, there is a typo in your caption for Figure 15 - it should read "(c)", not "(b)" after "NSIDC Age"*

AR: The typo is corrected.